# Towards Simple and Provable Parameter-Free Adaptive Gradient Methods

## Abstract

Optimization algorithms such as AdaGrad and Adam have significantly advanced the training of deep models by dynamically adjusting the learning rate during the optimization process. However, ad-hoc tuning of learning rates poses a challenge and leads to inefficiencies in practice. To address this issue, recent research has focused on developing "parameter-free" algorithms that operate effectively without the need for learning rate tuning. Despite these efforts, existing parameter-free variants of AdaGrad and Adam tend to be overly complex and/or lack formal convergence guarantees. In this paper, we present AdaGrad++ and Adam++, novel and simple parameter-free variants of AdaGrad and Adam with convergence guarantees. We prove that AdaGrad++ achieves comparable convergence rates to AdaGrad in convex optimization without predefined learning rate assumptions. Similarly, Adam++ matches the convergence rate of Adam without relying on any conditions on the learning rates. Experimental results across various deep learning tasks validate the competitive performance of Adam++.

## 1 Introduction

In recent years, optimization algorithms such as AdaGrad (Duchi et al., 2011) and Adam (Kingma, 2015) have emerged as powerful tools for enhancing the training of deep learning models by efficiently adapting the learning rate during the optimization process. While these algorithms have demonstrated remarkable performance gains in various applications, a notable drawback lies in the necessity of manual tuning for suitable learning rates. The process of learning rate tuning can be laborious and often requires extensive trial and error, hindering the efficiency and scalability of deep learning model training.

The intricate nature of learning rate tuning has motivated a large number of recent works to develop "learning-rate-free" or "parameter-free" algorithms that can work well under various different settings without learning rate tuning. Among the vast literature of parameter-free optimization methods, Ivgi et al. (2023) proposed a framework called distance over gradients (DoG), which gives a parameter-free version of stochastic gradient descent (SGD) that shares certain features with the AdaGrad-Norm algorithm (Streeter and McMahan, 2010; Ward et al., 2020). Motivated by AdaGrad-Norm, another recent work (Defazio and Mishchenko, 2023) also gave a framework named D-adaptation, and parameter-free variants of SGD and Adam were proposed under this framework. Defazio et al. (2024) proposed a different approach for schedule-free online optimization, based on which the authors developed new variants of schedule-free SGD and Adam/AdamW. Very recently, Kreisler et al. (2024) introduced a new parameter-free optimization algorithm named U-DoG, achieving a near-optimal convergence rate in smooth stochastic convex optimization by combining the adaptive learning rates introduced in Kavis et al. (2019) and Ivgi et al. (2023).

Despite the recent advances of parameter-free optimization algorithms, research on parameter-free adaptive gradient methods[1] remains relatively limited. Specifically, most existing parameter-free algorithms are variants of SGD, and *entry-wise adaptive learning rates* in standard AdaGrad and Adam algorithms are rarely considered in most of the existing parameter-free methods. Although

---

[1]Adaptive gradient methods usually have multiple hyperparameters other than learning rates. For example, Adam implements exponential moving averages of first and second moments of gradients, which are controlled by parameters $\beta_1$ and $\beta_2$. Here we clarify that when discussing parameter-free adaptive gradient methods, we still allow the algorithm to have such hyperparameters which do not require extensive tuning. This is consistent with the convention in recent works on parameter-free optimization (Defazio and Mishchenko, 2023; Mishchenko and Defazio, 2024; Defazio et al., 2024).

Table 1: Comparison of parameter-free (p.-f.) versions of AdaGrad and Adam and their convergence (conv.) guarantees in different works. In the table, we use the check mark (✓) to indicate that the corresponding paper gives the corresponding parameter-free algorithm or the convergence guarantee, while the cross mark (✗) indicates that the corresponding paper does not propose the corresponding algorithm or the convergence guarantee.

|  | p.-f. AdaGrad | conv. of p.-f. AdaGrad | p.f. Adam | conv. of p.-f. Adam |
|---|---|---|---|---|
| DoG (Ivgi et al., 2023) | ✗ | ✗ | ✗ | ✗ |
| D-adaptation (Defazio and Mishchenko, 2023) | ✓ | ✓ | ✓ | ✗ |
| Prodigy (Mishchenko and Defazio, 2024) | ✗ | ✗ | ✓ | ✗ |
| Schedule-Free (Defazio et al., 2024) | ✗ | ✗ | ✓ | ✗ |
| U-DoG (Kreisler et al., 2024) | ✗ | ✗ | ✗ | ✗ |
| This work | ✓ | ✓ | ✓ | ✓ |

Defazio and Mishchenko (2023); Mishchenko and Defazio (2024); Defazio et al. (2024) recently proposed variants of parameter-free AdaGrad, Adam and AdamW that implement entry-wise adaptive gradients, these algorithms all introduce rather significant modifications to the original algorithms, and the parameter-free versions of Adam/AdamW are not backed up by theoretical convergence guarantees.

Motivated by the limitations of existing studies, in this work, we propose simple but efficient versions of AdaGrad and Adam with provable convergence guarantees, which we name AdaGrad++ and Adam++[2] respectively. For ease of comparison, we summarize the results regarding parameter-free versions of AdaGrad and Adam in recent works in Table 1. The main contributions of this work can be summarized as follows:

- We propose the AdaGrad++ algorithm, which is a parameter-free version of AdaGrad. We demonstrate that without any assumptions on learning rates, AdaGrad++ can still achieve an $O(1/\sqrt{T})$ worst-case convergence rate in convex optimization, which is the same as AdaGrad. This highlights the efficacy and versatility of AdaGrad++ as a more accessible and user-friendly alternative.

- Based on AdaGrad++, we further derive the Adam++ algorithm as a parameter-free variant of Adam. By eliminating the reliance on a well-tuned learning rate schedule, Adam++ offers enhanced adaptability and robustness compared to Adam. Our theoretical results demonstrate the capability of Adam++ to match the convergence rate of Adam in convex optimization, even in the absence of any assumptions regarding learning rates.

- We conduct experiments on image classification and large language model pretraining tasks to evaluate the performance of the proposed algorithms. For CIFAR-10, with minimal parameter tuning, Adam++ outperforms Adam by 0.30% on accuracy using a cosine learning rate schedule on ResNet-16, and by 3.53% using a constant learning rate schedule on DenseNet-121. For GPT-2 small and medium tasks, Adam++ surpasses Adam by 0.02 in both training and test losses. Additionally, we perform an ablation study on the choice of initial and base learning rates, which confirms our theoretical findings.

**Notation.** We denote scalars by lowercase letters, vectors by lowercase boldface letters, and matrices by uppercase boldface letters. For a positive integer $d$, we denote $[d] = \{1, \ldots, d\}$. For a vector $\mathbf{x} = [x_1, \ldots, x_d]^\top$ and $p \geq 1$, we denote the $\ell_p$ norm of $\mathbf{x}$ by $\|\mathbf{x}\|_p = \left( \sum_{i=1}^{d} |x_i|^p \right)^{1/p}$, and the $\ell_\infty$ norm of $\mathbf{x}$ by $\|\mathbf{x}\|_\infty = \max_{i \in [d]} |x_i|$. Given two sequences $\{a_n\}$ and $\{b_n\}$, we write $a_n = O(b_n)$ if there exists a constant $C > 0$ such that $a_n \leq C\, b_n$. We use the notation $\widetilde{O}(\cdot)$ to hide logarithmic factors.

## 2 RELATED WORK

In this section, we give a more comprehensive review of the existing literature on parameter-free optimization and adaptive gradient methods.

---

[2]Adam++ follows certain designs and corrections proposed in the AMSGrad algorithm (Reddi et al., 2018). However, since AMSGrad is widely considered as an algorithm in the Adam family, we still name the algorithm as Adam++.

**Parameter-free optimization.** Several recent works have explored parameter-free algorithms based on modifications of the Polyak step size (Loizou et al., 2021; Gower et al., 2021; Orvieto et al., 2022; Rolinek and Martius, 2018; Berrada et al., 2020). In addition, several studies have investigated step-size selection methods derived from Line-Search algorithms (Vaswani et al., 2019; Paquette and Scheinberg, 2018). Another line of works, including LARS (You et al., 2017a), LAMB (You et al., 2017b), Adafactor (Simonyan and Zisserman, 2015), and Fromage (Bernstein et al., 2020), introduced learning rate adjustment schemes based on the norms of iterates. Moreover, Chandra et al. (2022) proposed a scheme to adjust the learning rates based on certain automatically calculated hypergradients. Several recent works (Orabona and Tommasi, 2017; Chen et al., 2022) have also proposed parameter-free algorithms by reducing the optimization process to a game of betting on a coin. Another recent work (Kleinsorge et al., 2023) proposed a novel rotation invariant parameter-free algorithm based on exponential learning rate adaption. Finally, a line of recent works (Orabona, 2014; Kempka et al., 2019) have studied parameter-free algorithms in solving specific learning tasks such as linear and kernel regression.

**Adaptive gradient methods.** There is a large body of work on variants of AdaGrad and Adam. Specifically, RMSProp (Kurbiel and Khaleghian, 2017) was the first work that proposed using an exponential moving average instead of a cumulative sum to handle the second moment in AdaGrad. Reddi et al. (2018) pointed out an extreme case where Adam may face convergence issues, and proposed AMSGrad accordingly with convergence guarantees. RMSProp, Adam and AMSGrad have also inspired many variants, including SC-AdaGrad, SC-RMSprop (Mukkamala and Hein, 2017), Sadagrad (Chen et al., 2018), YOGI (Zaheer et al., 2018), Padam (Chen et al., 2020), and RAdam (Liu et al., 2019). More recently, several works such as STORM (Cutkosky and Orabona, 2019), adaptive normalized SGD (Cutkosky and Mehta, 2020), Adam+ (Liu et al., 2020), SUPER-ADAM Huang et al. (2021) implemented various variance reduction techniques in Adam. Guo et al. (2021) presented a novel convergence analysis for a family of Adam-style methods with an increasing momentum parameter for the first-order moment. Alacaoglu et al. (2020) proposed a new type of framework to analyze the regret of the Adam style methods. Zhou et al. (2018) established high-probability convergence guarantees of AdaGrad and Adam in nonconvex optimization. Moreover, Taniguchi et al. (2024) proposed a new adaptive gradient method, ADOPT, which resolves Adam's non-convergence issue by removing the current gradient from the second-moment estimate and reordering the momentum and normalization updates. They also proved an $O(1/\sqrt{T})$ convergence rate with any choice of $\beta_2$ without depending on the bounded noise assumption.

## 3    REVIEW OF EXISTING METHODS AND PREVIEW OF PROPOSED METHODS

In this section, we give a brief review of the adaptive gradient methods, and discuss existing literature of parameter-free adaptive gradient methods, followed by a preview of our proposed methods.

We consider the optimization problem as follows

$$\min_{\mathbf{x} \in \mathbb{R}^d} f(\mathbf{x}), \tag{3.1}$$

where $f$ can be a convex or nonconvex function. In order to optimize (3.1), the standard stochastic gradient descent (SGD) performs the following update rule

$$\mathbf{x}_{t+1} = \mathbf{x}_t - \eta_t \mathbf{g}_t, \tag{3.2}$$

where $\mathbf{g}_t$ represents the stochastic gradient at the $t$-th iteration, $\eta_t$ denotes the learning rate. Adaptive gradient methods (Duchi et al., 2011; Hinton et al., 2012; Kingma, 2015; Reddi et al., 2018; Loshchilov and Hutter, 2019; Chen et al., 2020) aim to give well-designed adjustments to the learning rate $\eta_t$, particularly focusing on applying different learning rates for different entries of the iterates.

Among popular adaptive gradient methods, AdaGrad (Duchi et al., 2011) stands out as one of the pioneering methods. The update rule for AdaGrad is given by:

$$\mathbf{x}_{t+1} = \mathbf{x}_t - \frac{\eta_t}{\sqrt{\sum_{i=1}^t \mathbf{g}_i^2} + \delta} \cdot \mathbf{g}_t, \tag{3.3}$$

where $\delta$ is a small positive constant, and we use the common notation where the square $(\cdot)^2$ and square root $\sqrt{\cdot}$ operations are performed entry-wise when applied to a vector.

Adam (Kingma, 2015) is probably the most widely recognized adaptive gradient method. Compared with AdaGrad, it implements exponential moving averages over $\mathbf{g}_t^2$'s, as well as momentum

acceleration, with the update rule defined as follows:

$$\mathbf{x}_{t+1} = \mathbf{x}_t - \eta_t \frac{\mathbf{m}_t}{\sqrt{\mathbf{v}_t} + \delta}, \quad \mathbf{m}_t = \beta_1 \mathbf{m}_{t-1} + (1 - \beta_1)\mathbf{g}_t, \quad \mathbf{v}_t = \beta_2 \mathbf{v}_{t-1} + (1 - \beta_2)\mathbf{g}_t^2. \quad (3.4)$$

Another line of research on parameter-free optimization seeks to reduce or remove the necessity of learning rate tuning. The distance over gradient (DoG) (Ivgi et al., 2023) framework is a popular method which sets the learning rate $\eta_t$ in stochastic gradient descent (3.2) as

$$\eta_t = \frac{\max_{i \leq t} \|\mathbf{x}_0 - \mathbf{x}_i\|_2}{\sqrt{\sum_{i=1}^t \|\mathbf{g}_i\|_2^2}}.$$

DoG can be treated as a modification on the AdaGrad-Norm algorithm (Duchi et al., 2011; Streeter and McMahan, 2010; Ward et al., 2020) with $\eta_t = D/\sqrt{\sum_{i=1}^t \|\mathbf{g}_i\|_2^2}$, where the parameter $D$ is set as $\max_{i \leq t} \|\mathbf{x}_0 - \mathbf{x}_i\|_2$ in DoG. Several other parameter-free methods (Defazio and Mishchenko, 2023; Mishchenko and Defazio, 2024) also focused on estimating the parameter $D$ with different criteria. Notably, these recent studies of parameter-free algorithms focus more on the variants of SGD, which do not implement the entry-wise adaptive learning rates in AdaGrad and Adam. Although several recent works (Defazio and Mishchenko, 2023; Mishchenko and Defazio, 2024; Defazio et al., 2024) proposed parameter-free variants of AdaGrad or Adam, they are mostly not backed up with theoretical guarantees. Moreover, existing parameter-free variants of AdaGrad and Adam are mostly relatively complicated, deviating significantly from the standard forms of AdaGrad and Adam.

**Preview of our proposed methods.** Inspired by DoG (Ivgi et al., 2023), we propose simple parameter-free variants of AdaGrad and Adam, which we call AdaGrad++ and Adam++ respectively. Specifically, AdaGrad++ follows the update rule of AdaGrad in (3.3), but with

$$\eta_t = d^{-1/2} \cdot \max_{i \leq t} \|\mathbf{x}_i - \mathbf{x}_0\|_2,$$

where $d$ is the dimension of $\mathbf{x}$. Note that $\eta_t$ is the maximum distance between the initialization $\mathbf{x}_0$ and all the iterates along the optimization trajectory normalized by $\sqrt{d}$. Moreover, a specific and simplified case in Adam++ is directly based on the update rule of Adam in (3.4), with

$$\eta_t = \frac{\max_{i \leq t} \|\mathbf{x}_i - \mathbf{x}_0\|_2}{\sqrt{d(t+1)}}.$$

Compared with existing parameter-free versions of AdaGrad and Adam, AdaGrad++ and Adam++ are in a much simpler form. Interestingly, despite the simplicity, our analysis demonstrates that AdaGrad++ and Adam++ enjoy good theoretical convergence guarantees, and perform very well in various experiments. For more details, please refer to Sections 4 and 5.

## 4 ADAGRAD++: A PARAMETER-FREE VERSION OF ADAGRAD

In this section, we present the details of the AdaGrad++ algorithm, and then give theoretical guarantees on its performance in convex optimization.

### 4.1 ALGORITHM

We consider the optimization problem as introduced in (3.1) in the setting of stochastic optimization, and we assume access to a *stochastic gradient oracle* $\mathcal{G}(\mathbf{x})$ satisfying $\mathbb{E}[\mathcal{G}(\mathbf{x})|\mathbf{x}] \in \partial f(\mathbf{x})$. The AdaGrad++ algorithm is presented in Algorithm 1.

In Algorithm 1, it is clear that the key innovation of AdaGrad++ lies in the introduction of the quantity $r_t = \|\mathbf{x}_t - \mathbf{x}_0\|_2/\sqrt{d}$, and the definition that $\eta_t = \max(\eta_{t-1}, r_t)$. These definitions are inspired by the DoG framework (Ivgi et al., 2023), and are the key to a parameter-free approach. We would also like to comment that introducing the factor $d^{-1/2}$ in the definition of $r_t$ is crucial in AdaGrad++, resulting in strong theoretical guarantees and robust practical performance across different tasks with varying dimensions. The intuition is that AdaGrad++ implements different adaptive learning rates for different coordinates, and the $d^{-1/2}$ factor converts the "total distance" in DoG to the "mean squared distance (displacement)", which is more robust to $d$.

---

**Algorithm 1** Parameter-Free AdaGrad (AdaGrad++)

---

1: **input:** $\mathbf{x}_0, \eta_0 = \epsilon, \delta$
2: **for** $t = 0$, **to** $T$ **do**
3:    $r_t = \|\mathbf{x}_t - \mathbf{x}_0\|_2 / \sqrt{d}$
4:    $\eta_t = \max(\eta_{t-1}, r_t)$
5:    $\mathbf{g}_t = \mathcal{G}(\mathbf{x}_t)$
6:    $\mathbf{s}_t = (\sum_{k=0}^t \mathbf{g}_k \odot \mathbf{g}_i)^{1/2}$
7:    $\mathbf{H}_t = \delta + \mathrm{diag}(\mathbf{s}_t)$
8:    $\mathbf{x}_{t+1} = \mathbf{x}_t - \eta_t \cdot \mathbf{H}_t^{-1} \mathbf{g}_t$
9: **end for**

---

### 4.2 Convergence Guarantee

In this subsection, we present convergence guarantees of AdaGrad++ (Algorithm 1) under the setting where $f(\mathbf{x})$ is convex.[3] We first give an assumption on the stochastic gradient $\mathcal{G}(\mathbf{x})$.

**Assumption 4.1.** There exists some continuous function $l : \mathbb{R}^d \to \mathbb{R}$ such that $\|\mathcal{G}(\mathbf{x})\|_2 \leq l(\mathbf{x})$ almost surely.

Assumption 4.1 states that the stochastic gradients have a deterministic bound $l(\mathbf{x})$ on their norm. By allowing different bounds at different $\mathbf{x}$, this assumption is much weaker compared to the more common Lipschitz assumption that directly requires that $\|\mathcal{G}(\mathbf{x})\|_2$ is bounded to a constant. The same assumption has been made in Ivgi et al. (2023). Note that this assumption is strictly weaker than the typical bounded gradient norm assumption, and as we will show later, it yields stronger convergence results.

Our main result on the convergence of AdaGrad++ is given in the following theorem.

**Theorem 4.2.** Let $\mathbf{x}_0, \ldots, \mathbf{x}_T$ be the iterates of AdaGrad++. Further let $\tau \in \arg\max_{t \leq T} \sum_{i=0}^{t-1} \frac{\eta_i}{\eta_t}$ and define $\overline{\mathbf{x}}_\tau = \frac{\sum_{t=0}^{\tau-1} \eta_t \mathbf{x}_t}{\sum_{t=0}^{\tau-1} \eta_t}$. Then under Assumption 4.1, for any $\delta \in (0, 1)$, $L > 0$ and any $\mathbf{x}^* \in \mathbb{R}^d$, with probability at least $1 - \delta - \mathbb{P}(\max_{t \leq T} l(\mathbf{x}_t) > L)$, it holds that

$$f(\overline{\mathbf{x}}_\tau) - f(\mathbf{x}^*) \leq O\left( \frac{M_1 \|\mathbf{s}_\tau\|_2 + \sqrt{M_2 \|\mathbf{s}_\tau\|_2^2 + M_3}}{T} \right),$$

where

$$M_1 = \frac{D_\tau^2 \sqrt{d}}{\eta_0} \cdot \log\left( \frac{\eta_T}{\eta_0} \right), \; M_2 = \overline{D}_\tau^2 \log^2\left( \frac{\eta_T}{\eta_0} \right) \cdot \log\left[ \frac{60 \log(6t)}{\delta} \right],$$

$$M_3 = L^2 \log^2\left( \frac{\eta_T}{\eta_0} \right) \cdot \log^2\left[ \frac{60 \log(6t)}{\delta} \right],$$

where $D_\tau = \max_{t \leq \tau} \|\mathbf{x}_t - \mathbf{x}^*\|_\infty$, $\overline{D}_\tau = \max_{t \leq \tau} \|\mathbf{x}_t - \mathbf{x}^*\|_2$.

Theorem 4.2 gives the bound of $f(\overline{\mathbf{x}}_\tau)$ that is defined by an arbitrarily chosen reference point $\mathbf{x}^*$, and the bound contains a term $f(\mathbf{x}^*)$ as well as several other terms that are related to the distance between algorithm iterates and $\mathbf{x}^*$. This type of bound matches standard bounds in convex and Lipschitz/smooth optimization (Bubeck et al., 2015). Moreover, the probability for the bound in Theorem 4.2 to hold depends on $\mathbb{P}(\max_{t \leq T} l(\mathbf{x}_t) > L)$, and the bound holds with high probability when $\mathbb{P}(\max_{t \leq T} l(\mathbf{x}_t) > L)$ is small. It is worth noting that if $l(\cdot)$ is always bounded, which corresponds to a Lipschitz $f$, then $\mathbb{P}(\max_{t \leq T} l(\mathbf{x}_t) > L) = 0$ with an appropriately chosen constant $L$. Notably, Theorem 4.2 covers more general and non-Lipschitz cases as well, since $l(\cdot)$ only needs to be bounded along the optimization trajectory $\mathbf{x}_0, \ldots, \mathbf{x}_T$ to grant $\mathbb{P}(\max_{t \leq T} l(\mathbf{x}_t) > L) = 0$.

In addition, it is worth noting that $D_\tau = \max_{t \leq \tau} \|\mathbf{x}_t - \mathbf{x}^*\|_\infty$ and $\bar{D}_\tau = \max_{t \leq \tau} \|\mathbf{x}_t - \mathbf{x}^*\|_2$ depends on $t$ and may grow as $t$ increases in the worst case. The same issue appears in DoG as discussed in Ivgi et al. (2023). Nevertheless, Ivgi et al. (2023) argue that $\|\mathbf{x}_t - \mathbf{x}^*\|_2$ is typically bounded by $O(\|\mathbf{x}_0 - \mathbf{x}^*\|_2)$ for all $t$, e.g., $\|\mathbf{x}_t - \mathbf{x}^*\|_2 \leq 3\|\mathbf{x}_0 - \mathbf{x}^*\|_2$, so the iterates do not move too far away

---

[3]We also provide convergence guarantees for AdaGrad++ in the nonconvex setting in Appendix D.

---

**Algorithm 2** Parameter-Free Adam (Adam++)

1: **input:** $\mathbf{x}_0, \eta_0 = \epsilon, \delta, \beta_1, \beta_2, \lambda$
2: **for** $t = 0$, **to** $T$ **do**
3:    $r_t = \|\mathbf{x}_t - \mathbf{x}_0\|_2 / \sqrt{d}$
4:    $\eta_t = \max(\eta_{t-1}, r_t)$
5:    $\mathbf{g}_t = \mathcal{G}(\mathbf{x}_t)$
6:    $\beta_{1t} = \beta_1 \lambda^{t-1}$
7:    $\mathbf{m}_t = \beta_{1t} \mathbf{m}_{t-1} + (1 - \beta_{1t}) \mathbf{g}_t$
8:    **Case 1:** $\mathbf{s}_t = (\sum_{k=0}^{t} \mathbf{g}_k \odot \mathbf{g}_k)^{1/2}$
9:    **Case 2:** $\mathbf{v}_t = \beta_2 \mathbf{v}_{t-1} + (1 - \beta_2) \mathbf{g}_t \odot \mathbf{g}_t, \mathbf{s}_t = \sqrt{(t+1) \cdot \max_{k \leq t}(\mathbf{v}_k)}$
10:    $\mathbf{H}_t = \delta + \mathrm{diag}(\mathbf{s}_t)$
11:    $\mathbf{x}_{t+1} = \mathbf{x}_t - \eta_t \cdot \mathbf{H}_t^{-1} \mathbf{m}_t$
12: **end for**

---

from $\mathbf{x}_0$. This is also observed in our experiments. In addition, Ivgi et al. (2023) proposed a step size scheme whose iterates are guaranteed to satisfy $\|\mathbf{x}_t - \mathbf{x}^*\|_2 \leq O(\|\mathbf{x}_0 - \mathbf{x}^*\|_2)$ with high probability (See Section 3.3 and Proposition 2 in their paper). By using a similar step size scheme, we can also develop modified versions of AdaGrad++ (and Adam++ in the next section), such that the distance $\|\mathbf{x}_t - \mathbf{x}^*\|_2$ will be bounded by $O(\|\mathbf{x}_0 - \mathbf{x}^*\|_2)$ with high probability. Therefore, in the following discussion, we assume $\eta_t \leq O(\|\mathbf{x}_0 - \mathbf{x}^*\|_2/\sqrt{d})$, $D_\tau = \max_{t \leq \tau} \|\mathbf{x}_t - \mathbf{x}^*\|_\infty \leq O(\|\mathbf{x}_0 - \mathbf{x}^*\|_\infty)$ and $\bar{D}_\tau = \max_{t \leq \tau} \|\mathbf{x}_t - \mathbf{x}^*\|_2 \leq O(\|\mathbf{x}_0 - \mathbf{x}^*\|_2)$. Under these assumptions, $M_1$, $M_2$ and $M_3$ can be viewed as some constants up to logarithmic factors.

With these assumptions in hand, Theorem 4.2 reveals that an important term $\|\mathbf{s}_\tau\|_2$ determines the convergence rate of AdaGrad++. We note that a similar quantity has been investigated by Zhou et al. (2018) in the study of nonconvex convergence guarantees of adaptive gradient methods. This similarity demonstrates that our proposed parameter-free algorithm AdaGrad++ still captures the key nature of AdaGrad. Taking a closer look at the quantity $\|\mathbf{s}_\tau\|_2$, by definition, we have $\|\mathbf{s}_\tau\|_2 = \sqrt{\sum_{t=0}^{\tau} \|\mathbf{g}_t\|_2^2}$. When the objective function is Lipschitz (i.e., $l(\cdot)$ is bounded), it is clear that a worst-case upper bound of $\|\mathbf{s}_\tau\|_2$ is $\sqrt{T}$, leading to a $1/\sqrt{T}$ bound on the convergence rate (see Corollary A.1 in Appendix A). However, as discussed in Zhou et al. (2018), here we point out that in practice, we often observe that $\|\mathbf{s}_\tau\|_2 \ll \sqrt{T}$ due to the fact that the algorithm converges and the stochastic gradients $\|\mathbf{g}_t\|_2$ may converge to zero. When $\|\mathbf{s}_\tau\|_2 = O(T^{1/2-\alpha})$ for some $\alpha \in (0, 1/2)$, we will have a better convergence rate of AdaGrad++ (see Corollary A.2 in Appendix A).

Finally, $M_1$ has a $\sqrt{d}$ dependence, which may seem large. However, this is not the case, since $D_\tau$ is defined with respect to the vector infinity norm, and thus $D_\tau^2 \sqrt{d}$ is of the same order as $\bar{D}_\tau^2$.

## 5 ADAM++: A PARAMETER-FREE VERSION OF ADAM

Based on the analysis of Adagrad++, in this section, we introduce the Adam++ algorithm together with its theoretical convergence guarantees.

### 5.1 ALGORITHM

We consider the same optimization problem as introduced in (3.1) in the stochastic setting. We also consider the same stochastic gradient oracle $\mathcal{G}(\mathbf{x})$ satisfying $\mathbb{E}[\mathcal{G}(\mathbf{x})|\mathbf{x}] \in \partial f(\mathbf{x})$. The Adam++ algorithm is depicted in Algorithm 2. There are several key points in Algorithm 2 to note. First of all, Adam++ also implements the key quantity $r_t = \|\mathbf{x}_t - \mathbf{x}_0\|_2/\sqrt{d}$ introduced in AdaGrad++ to automatically adapt the "learning rate". Moreover, Adam++ allows dynamically decaying first-moment parameter $\beta_{1t} = \beta_1 \lambda^t$, which follows the definition in AMSGrad (Reddi et al., 2018). When setting $\lambda = 1$, we can recover the common setup with a constant $\beta_1$. The introduction of the decaying $\beta_{1t}$ is due to technical reasons, and our theoretical analysis on Adam relies on a $\lambda \in (0, 1)$. However, we remark that Adam++ with $\lambda = 1$ can achieve highly competitive performance under various practical settings.

Another key feature of Adam++ is that it covers two cases. In **Case 1**, we implement entry-wise adaptive learning rates that are similar to AdaGrad and AdaGrad++. In **Case 2**, we implement a more common exponential moving average of the second moment $\mathbf{v}_t$ but also introduce another

quantity $\mathbf{s}_t$. Particularly regarding the definition of $\mathbf{s}_t = \sqrt{(t+1) \cdot \max_{t' \leq t}(\mathbf{v}_{t'})}$, we note that the factor $\sqrt{(t+1)}$ ensures reasonable scaling when incorporated with the quantity $r_t$. This factor makes the scaling of $\mathbf{s}_t$ in **Case 2** more compatible with that in **Case 1**. Moreover, the max operation $\max_{t' \leq t}(\mathbf{v}_{t'})$ is inherited from the AMSGrad modification to Adam (Reddi et al., 2018), which has been shown to be crucial for establishing convergence guarantees. However, in practice, Adam (or AdamW) performs better than AMSGrad for training LLMs. Therefore, we apply Adam-like update rule $\mathbf{s}_t = \sqrt{(t+1)\mathbf{v}_t}$ for language modeling tasks and use AMSGrad-like update rule (Algorithm 2 Case 2) for other computer vision experiments.

## 5.2 Convergence Guarantee of Adam++

In this section, we give the convergence guarantee of Adam++. Our result will be based on the same assumption (Assumption 4.1) as that of AdaGrad++. The main result is given in the following theorem.

**Theorem 5.1.** Let $\mathbf{x}_0, \ldots, \mathbf{x}_T$ be the iterates of Adam++ following either **Case 1** or **Case 2** in Algorithm 2. In addition, let $\tau \in \arg\max_{t \leq T} \sum_{i=0}^{t-1} \frac{\eta_i}{\eta_t}$ and define $\overline{\mathbf{x}}_\tau = \frac{\sum_{t=0}^{T-1} \eta_t \mathbf{x}_\tau}{\sum_{t=0}^{T-1} \eta_t}$. Suppose $0 < \beta_1 < \sqrt{\beta_2}$ and $0 < \lambda < 1$. Then under Assumption 4.1, for any $\delta \in (0, 1)$, $L > 0$ and any $\mathbf{x}^* \in \mathbb{R}^d$, with probability at least $1 - \delta - \mathbb{P}(\max_{t \leq T} l(\mathbf{x}_t) > L)$, the following results hold:

$$f(\overline{\mathbf{x}}_\tau) - f(\mathbf{x}^*) \leq O\left( \frac{M_1 \|\mathbf{s}_\tau\|_2 + \sqrt{M_2 \|\mathbf{s}_\tau\|_2^2 + M_3}}{T} \right),$$

Here, $D_\tau = \max_{t \leq \tau} \|\mathbf{x}_t - \mathbf{x}^*\|_\infty$, $\overline{D}_\tau = \max_{t \leq \tau} \|\mathbf{x}_t - \mathbf{x}_*\|_2$, and

$$M_1 = \frac{D_\tau^2 \sqrt{d}}{\eta_0} \cdot \log\left(\frac{\eta_T}{\eta_0}\right), \quad M_2 = \overline{D}_\tau^2 \log^2\left(\frac{\eta_T}{\eta_0}\right) \cdot \log\left[\frac{60 \log(6t)}{\delta}\right],$$

$$M_3 = L^2 \log^2\left(\frac{\eta_T}{\eta_0}\right) \cdot \log^2\left[\frac{60 \log(6t)}{\delta}\right],$$

Theorem 5.1 gives the convergence guarantee for Adam++. To the best of our knowledge, this is the first convergence guarantee of a parameter-free version of Adam. The bound holds with high probability when $l(\cdot)$ is bounded along the optimization trajectory $\mathbf{x}_0, \ldots, \mathbf{x}_T$. As we explained before, it is reasonable to assume that $M_1$, $M_2$ and $M_3$ are constants up to logarithmic factors. The subsequent discussion proceeds under this assumption. Similar to the bound for AdaGrad++, the quantity $\|\mathbf{s}_\tau\|_2$ is a key quantity: when $l(\mathbf{x})$ is bounded, the worst-case bound of $\|\mathbf{s}_\tau\|_2$ is $O(\sqrt{T})$, leading to a $\widetilde{O}(1/\sqrt{T})$ convergence rate. However, if $\|\mathbf{s}_\tau\|_2 = O(T^{1/2-\alpha})$ for some $\alpha \in (0, 1/2)$, we can expect a faster convergence rate.

Clearly, we can also establish the counterparts of Corollaries A.1 and A.2 for Adam++. However, to avoid repetitions, here we only give the corollary below as the counterpart of Corollary A.2. The counterpart of Corollary A.1 can be obtained by setting $\alpha = 0$.

**Corollary 5.2.** Suppose that the assumptions in Theorem 5.1 hold. Further assume that there exist $G > 0$ such that $l(\mathbf{x}) \leq G$ and $\|\mathbf{s}_\tau\|_2 \leq G \cdot T^{1/2-\alpha}$ for some $\alpha \in [0, 1/2)$. Then for any $\mathbf{x}^* \in \mathbb{R}^d$, with probability at least $1 - \delta$, it holds that

$$f(\overline{\mathbf{x}}_\tau) - f(\mathbf{x}_*) \leq \widetilde{O}\left( \frac{D_\tau^2 G \cdot \sqrt{d}}{T^{1/2+\alpha}} \right),$$

where $D_\tau = \max_{t \leq \tau} \|\mathbf{x}_t - \mathbf{x}_*\|_\infty$.

In the worst case where $\alpha = 0$, Corollary 5.2 gives a convergence rate of order $\sqrt{d/T}$ for Adam++. This matches the convergence rate in the original paper of AMSGrad (Theorem 4 and Corollary 1 in (Reddi et al., 2018)) as well as those in more recent works (e.g., Theorems 4.3, 5.2, Corollaries 4.6, 5.5 in Zhou et al. (2018) and Theorems 1,2,3,4 in Défossez et al.). We note that a very recent work (Ahn and Cutkosky, 2024) gives a new convergence result for Adam improving the dependency in the dimension $d$. However, Ahn and Cutkosky (2024) focuses on the nonconvex setting and studies the convergence rate towards a stationary point, which is different from the setting considered here. We will also provide the convergence guarantee of Adam++ for nonconvex optimization in Appendix D.

## 6 EXPERIMENTS

In this section, we evaluate the performance of Adam++ across image classification and large language model pretraining tasks to test its efficacy. For image classification problems, we train models on the CIFAR-10 dataset and CIFAR-100 datasets (Krizhevsky et al., 2009). To demonstrate Adam++'s versatility and stability across different network structures, we apply it to neural network architectures including ResNet-18 (He et al., 2016), VGG16 (Simonyan and Zisserman, 2015), and DenseNet-121 (Huang et al., 2017). We use AdamW, a version of Adam (Kingma, 2015) with decoupled weight decay[4], as the baseline, and also compare Adam++ against two state-of-the-art parameter-free algorithms: D-Adaptation Adam (Defazio and Mishchenko, 2023) and Prodigy (Mishchenko and Defazio, 2024). For large language model pretraining tasks, we use a reproduced GPT-2 model with 125M and 355M parameters respectively on the OpenWebText dataset (Gokaslan and Cohen, 2019). Our training settings are based on those from NanoGPT and Sophia (Liu et al., 2023). We just present some experiments of image classification task here and postpone other experiments including large language model experiments and ablation study to Appendix H due to page limitation. We omit the experiments for AdaGrad++ as we found it consistently underperforms compared to Adam and Adam++, despite being better than AdaGrad. The code is available at `https://anonymous.4open.science/r/Adampp_pub-1626`.

### 6.1 IMAGE CLASSIFICATION

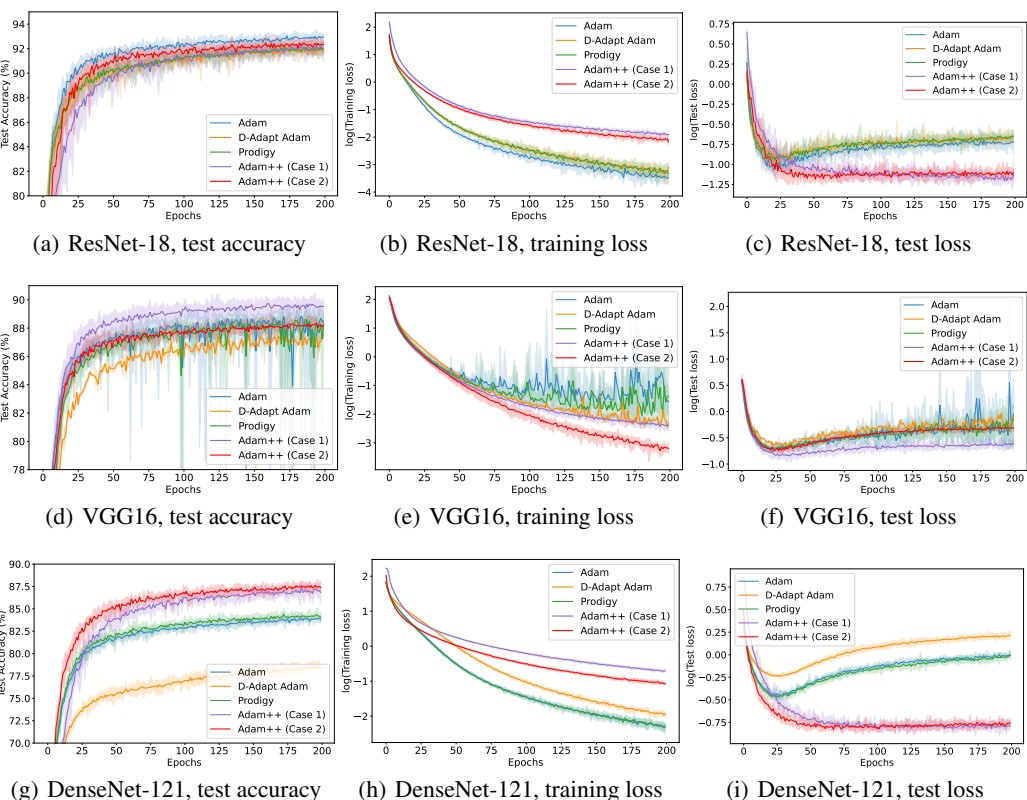

(a) ResNet-18, test accuracy    (b) ResNet-18, training loss    (c) ResNet-18, test loss

(d) VGG16, test accuracy    (e) VGG16, training loss    (f) VGG16, test loss

(g) DenseNet-121, test accuracy    (h) DenseNet-121, training loss    (i) DenseNet-121, test loss

Figure 1: The results of training ResNet-18, VGG16 and DenseNet-121 on CIFAR-10 with a constant learning rate schedule.

We aim to compare the optimization algorithms in a setting with minimal or no parameter tuning. On ResNet-18, VGG16 and DenseNet-121, we run the baseline AdamW optimizer with learning rate searched from $1 \times 10^{-4}$ to $1 \times 10^{-2}$ and a decoupled weight decay of $5 \times 10^{-4}$. For all parameter-free algorithms, including DAdapt Adam, Prodigy, and Adam++, although there is no learning rate choice required, we set a base learning rate factor that can be applied on top of the adaptive learning rate, as

---

[4]To compare with D-Adaptation Adam, we still denote "Adam" in the figures.

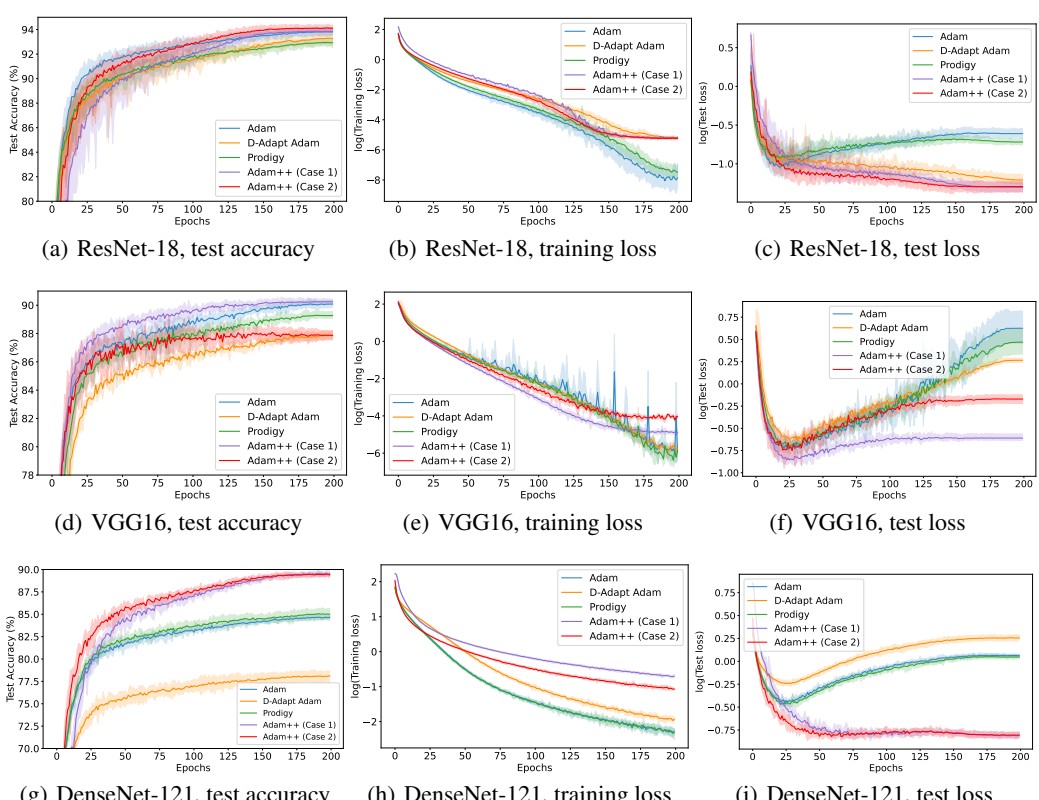

Figure 2: The results of training ResNet-18, VGG16 and DenseNet-121 on CIFAR-10 with a cosine learning rate schedule.

introduced in Ivgi et al. (2023); Mishchenko and Defazio (2024); Defazio and Mishchenko (2023). For these parameter-free optimizers, we search for the base learning rate factor ranging from 0.1 to 3.0, while keeping all other parameters consistent with those of AdamW, ensuring a fair comparison. [5]. Moreover, we repeat these experiments for 8 times with different random seeds to reduce the effect of randomness, and we plot the range of these curves with colored shadows. For model architectures, we modify the output dimensions of these networks to 10 in CIFAR-10 to align with the number of output classes. We provide a detailed list of all training parameters in Appendix G. And the results demonstrate that our algorithm demonstrates superior performances to other algorithms.

**Constant Learning Rate Schedule** Figure 1 illustrates the training loss, test loss and test accuracy curves against training epochs on the CIFAR-10 dataset for various network architectures and algorithms. The task is challenging due to the use of a fixed learning rate throughout all epochs. For the Adam++ algorithm, we implement both Case 1 and Case 2 variants. It can be observed that Adam++ demonstrates superior performance to baseline parameter-free optimization methods, including Prodigy and D-Adaptation Adam. Moreover, Adam++ either matches or surpasses Adam's performance. On DenseNet-121, Adam++ even achieves much higher test accuracies than AdamW.

**Cosine Learning Rate Schedule** In addition to the learning rates found by parameter-free algorithms, it is common to apply an additional learning rate schedule on top of that according to (Ivgi et al., 2023; Mishchenko and Defazio, 2024; Defazio and Mishchenko, 2023). Figure 2 provides a comparison of our algorithm with other baselines with cosine scheduler. This annealed schedule aids in stabilizing training by being more cautious near the optimal point. Under the annealed setting, all the algorithms exhibit improvement over their counterparts with a constant learning rate. Moreover, Adam++ can also achieve comparable or even better performance than AdamW in most cases, and maintain a competitive edge over other parameter-free algorithms including D-Adaptation Adam.

---

[5]We discuss the reason for learning rate search on parameter-free algorithms in Appendix F.

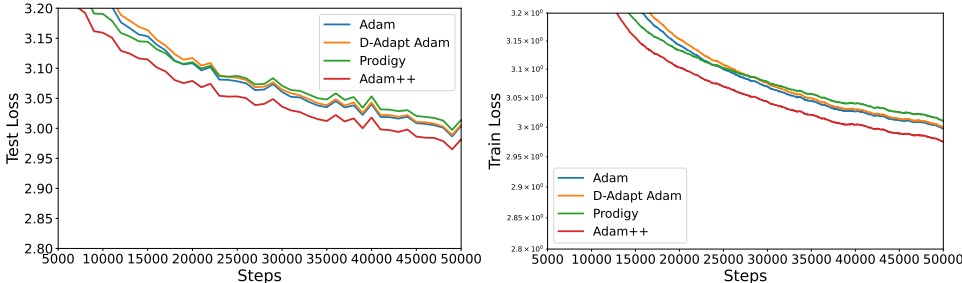

Figure 3: Comparison of training GPT-2 Small (155M) on OpenWebText. Left: Test loss. Performance at 50k steps—AdamW: 3.00, D-Adapt AdamW: 3.01, Prodigy: 3.01, Adam++: 2.98. Right: Train loss. Performance at 50k steps—AdamW: 2.97, D-Adapt AdamW: 2.97, Prodigy: 2.98, AdamW++: 2.95. AdamW++ refers to AdamW++ (Case 2).

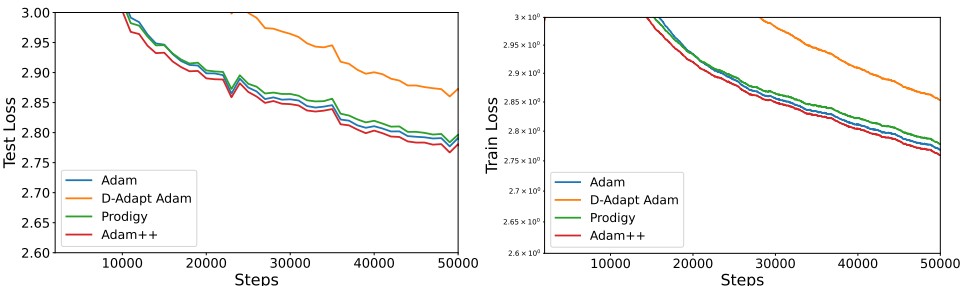

Figure 4: Comparison of training GPT-2 Medium (355M) on OpenWebText. Left: Test loss. Performance at 50k steps—AdamW: 2.80, D-Adapt AdamW: 2.87, Prodigy: 2.80, AdamW++: 2.78. Right: Train loss. Performance at 50k steps—AdamW: 2.75, D-Adapt AdamW: 2.82, Prodigy: 2.75, AdamW++: 2.73. AdamW++ refers to AdamW++ (Case 2).

## 6.2 LARGE LANGUAGE MODEL (LLM) PRETRAINING

In this subsection, we pretrain GPT-2 models with 125M and 355M parameters using the OpenWeb-Text dataset. For the baseline, we employ the AdamW optimizer instead of Adam, since empirically AdamW performs better than Adam in LLM tasks. For all parameter-free algorithms, including our proposed Adam++, we apply decoupled weight decay to align with AdamW, referring to the adjusted version of Adam++ as AdamW++. In detail, AdamW uses a standard cosine learning rate schedule with 2000 warm-up steps. The batch size is set to 480, with a learning rate of $6 \times 10^{-4}$ for GPT-2 small and $3 \times 10^{-4}$ for GPT-2 medium, as specified in Liu et al. (2023). All parameter-free algorithms use the same hyperparameters and learning rate schedule as AdamW. Additional details for pretraining are provided in Appendix G.

From Figures 3 and 4, we observe that AdamW++ outperforms AdamW by 0.02 in both training loss and validation loss on GPT-2 small and GPT-2 medium. In contrast, Prodigy is worse than AdamW on GPT-2 small and matches AdamW on GPT-2 medium, while D-Adapt Adam shows the weakest performance on these tasks. These results emphasize the ability of our algorithm to effectively handle large-scale language modeling tasks.

## 7 CONCLUSION AND FUTURE WORK

In this paper, we propose two simple but effective algorithms, namely AdaGrad++ and Adam++, that are parameter-free variants of AdaGrad and AdamW respectively. We demonstrate that, despite the simple intuition, AdaGrad++ and Adam++ are guaranteed to converge with a reasonable convergence rate, and also perform surprisingly well in various experiments. These theoretical and empirical results highlight the potential of AdaGrad++ and Adam++ to be robust and practical choices for a wide range of optimization tasks.

We note that the proposed method still requires tuning of the base learning rate, which is a common limitation of all existing parameter-free optimization algorithms. In future work, we aim to eliminate this requirement. Moreover, AdamW++ is used in our experiments without proof, and establishing convergence guarantees for AdamW++ is another promising area for future work.

## ETHICS STATEMENT AND LLM USAGE

This work introduces two kinds of parameter-free optimization algorithms that demonstrate better efficiency on training deep learning neural networks. These methods have the potential for accelerating innovation and application of AI techniques in downstream fields such as education and healthcare. However, latent threats to the job market and potential promotion for the spread of false information should be further investigated.

Our use of Large Language Models (LLMs) is limited to the GPT-2 experiments in Section 6.2.

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

# A  FURTHER DISCUSSION ON ADAGRAD++

Based on Theorem 4.2, we have the following corollaries:

**Corollary A.1.** Suppose that the assumptions in Theorem 4.2 hold. Further assume that $l(\mathbf{x}) \leq G$ for all $\mathbf{x}$. Then for any $\mathbf{x}^* \in \mathbb{R}^d$, with probability at least $1 - \delta$, it holds that

$$f(\overline{\mathbf{x}}_\tau) - f(\mathbf{x}^*) \leq \widetilde{O}\left( D_\tau^2 G \cdot \sqrt{\frac{d}{T}} \right),$$

where $D_\tau = \max_{t \leq \tau} \|\mathbf{x}_t - \mathbf{x}_*\|_\infty$.

Corollary A.1 gives a simplified version of Theorem 4.2 under the special case when $l(\mathbf{x}) \leq G$. We note that Mishchenko and Defazio (2024) proposed a parameter-free version of AdaGrad named D-Adapted AdaGrad and established a convergence rate of the order $O(dG_\infty/\sqrt{T})$, under the assumption that $\|\mathcal{G}(\mathbf{x})\|_\infty \leq G_\infty$. Considering $\|\mathcal{G}(\mathbf{x})\|_2 \leq \sqrt{d} \cdot \|\mathcal{G}(\mathbf{x})\|_\infty$, we have $G \leq \sqrt{d} \cdot G_\infty$, and therefore our result can be reduced to the bound in Mishchenko and Defazio (2024) when the distance factor $D_\tau$ is omitted.

**Corollary A.2.** Suppose that the assumptions in Theorem 4.2 hold. Further assume that there exist $G > 0$ such that $l(\mathbf{x}) \leq G$ and $\|\mathbf{s}_\tau\|_2 \leq G \cdot T^{1/2-\alpha}$ for some $\alpha \in [0, 1/2)$. Then for any $\mathbf{x}^* \in \mathbb{R}^d$, with probability at least $1 - \delta$, it holds that

$$f(\overline{\mathbf{x}}_\tau) - f(\mathbf{x}_*) \leq \widetilde{O}\left( \frac{D_\tau^2 G \cdot \sqrt{d}}{T^{1/2+\alpha}} \right),$$

where $D_\tau = \max_{t \leq \tau} \|\mathbf{x}_t - \mathbf{x}_*\|_\infty$.

Corollary A.2 is a straightforward simplification of Theorem 4.2 under the additional condition that $\|\mathbf{s}_\tau\|_2 \leq G \cdot T^{1/2-\alpha}$. It suggests that when the key quantity $\|\mathbf{s}_\tau\|_2$ is smaller than the worst-case $O(\sqrt{T})$ bound, the convergence rate can be faster than $O(1/\sqrt{T})$.

# B  PROOF OF THEOREM 4.2

Before we start the proof, we provide the necessary background and definitions that will be used in our proof.

**Definition B.1** (Bregman Divergence, Bregman 1967). Let $\psi : \Omega \to \mathbb{R}$ be a strictly convex and continuously differentiable function defined on a closed convex set $\Omega$. Then the Bregman divergence is defined as

$$B_\psi(\mathbf{x}, \mathbf{y}) = \psi(\mathbf{x}) - \psi(\mathbf{y}) - \langle \nabla\psi(\mathbf{y}), \mathbf{x} - \mathbf{y} \rangle, \forall \mathbf{x}, \mathbf{y} \in \Omega.$$

The Bregman distance is the difference between the value of $\psi$ at $\mathbf{x}$ and the value of the first-order Taylor expansion of $\psi$ around $\mathbf{y}$ evaluated at point $\mathbf{x}$.

From the definition, it is easy to verify the following properties of Bregman divergence:

- Gradient at $\mathbf{x}$: $\nabla_{\mathbf{x}} B_\psi(\mathbf{x}, \mathbf{y}) = \nabla\psi(\mathbf{x}) - \nabla\psi(\mathbf{y}), \forall \mathbf{x}, \mathbf{y} \in \Omega$.

- Three-points identity: $B_\psi(\mathbf{x}, \mathbf{y}) + B_\psi(\mathbf{y}, \mathbf{z}) - B_\psi(\mathbf{x}, \mathbf{z}) = \langle \nabla\psi(\mathbf{z}) - \nabla\psi(\mathbf{y}), \mathbf{x} - \mathbf{y} \rangle$, $\forall \mathbf{x}, \mathbf{y}, \mathbf{z} \in \Omega$.

*Proof of Theorem 4.2.* For step $k$, let $\mathbf{d}_k = \mathbf{x}_k - \mathbf{x}_*$, and $\psi_k(\mathbf{x}) = \langle \mathbf{x}, \mathbf{H}_k \mathbf{x} \rangle$. We define a Bregman divergence $B_{\psi_k}(\mathbf{x}, \mathbf{y}) = \psi_k(\mathbf{x}) - \psi_k(\mathbf{y}) - \langle \nabla\psi_k(\mathbf{y}), \mathbf{x} - \mathbf{y} \rangle = \psi_k(\mathbf{x} - \mathbf{y})/2$. Let $\mathbf{g}_t = (g_{k,1}, \cdots, g_{k,d})^\top, \mathbf{s}_k = (s_{k,1}, \cdots, s_{k,d})^\top$ and $\mathbf{d}_k = (d_{k,1}, \cdots, d_{k,d})^\top$. By the definition of $\mathbf{x}_{k+1}$, we have

$$\mathbf{x}_{k+1} = \arg\min_{\mathbf{x}} \{\eta_k \langle \mathbf{g}_k, \mathbf{x} \rangle + B_{\psi_k}(\mathbf{x}, \mathbf{x}_k)\}.$$

We have for all $\mathbf{x}$ that

$$\langle \mathbf{x} - \mathbf{x}_{k+1}, \eta_k \mathbf{g}_k + \nabla\psi_k(\mathbf{x}_{k+1}) - \nabla\psi_k(\mathbf{x}_k) \rangle \geq 0, \tag{B.1}$$

where we use the first-order optimality condition and the property of the gradient of Bregman divergence at $\mathbf{x}$. Setting $\mathbf{x} = \mathbf{x}^*$ and rearranging the terms, we can then obtain a bound of $\langle \mathbf{x}_{k+1} - \mathbf{x}^*, \mathbf{g}_k \rangle$. Thus we have the inequality by denoting the dual norm of $\|\cdot\|_{\psi_k}$ by $\|\cdot\|_{\psi_k^*}$:

$$
\begin{aligned}
\eta_k \langle \mathbf{x}_k - \mathbf{x}^*, \mathbf{g}_k \rangle &= \eta_k \langle \mathbf{x}_{k+1} - \mathbf{x}^*, \mathbf{g}_k \rangle + \eta_k \langle \mathbf{x}_k - \mathbf{x}_{k+1}, \mathbf{g}_k \rangle \\
&\leq \langle \mathbf{x}^* - \mathbf{x}_{k+1}, \nabla \psi_k(\mathbf{x}_{k+1}) - \nabla \psi_k(\mathbf{x}_k) \rangle + \frac{1}{2} \psi_k(\mathbf{x}_{k+1} - \mathbf{x}_k) + \frac{1}{2} \psi_k^*(\eta_k \mathbf{g}_k) \\
&= \langle \mathbf{x}^* - \mathbf{x}_{k+1}, \nabla \psi_k(\mathbf{x}_{k+1}) - \nabla \psi_k(\mathbf{x}_k) \rangle + B_{\psi_k}(\mathbf{x}_{k+1}, \mathbf{x}_k) + \frac{\eta_k^2}{2} \|\mathbf{g}_k\|_{\psi_k^*}^2 \\
&= B_{\psi_k}(\mathbf{x}^*, \mathbf{x}_k) - B_{\psi_k}(\mathbf{x}^*, \mathbf{x}_{k+1}) + \frac{\eta_k^2}{2} \|\mathbf{g}_k\|_{\psi_k^*}^2, \quad\quad (B.2)
\end{aligned}
$$

where the inequality follows from (B.1) and the Cauchy-Schwarz inequality for $\langle \mathbf{x}_k - \mathbf{x}_{k+1}, \eta_k \mathbf{g}_k \rangle$, the second equality follows from the definition of $\psi_k$ and $B_{\psi_k}$, and the last equality follows from Three-points identity of Bregman divergence. In addition, we define $\overline{\mathbf{x}}_t := \frac{1}{\sum_{k=0}^{t-1} \eta_k} \sum_{k=0}^{t-1} \eta_k \mathbf{x}_k$ and $\Delta_t := \nabla f(\mathbf{x}_t) - \mathbf{g}_t$, then we have

$$
\begin{aligned}
f(\overline{\mathbf{x}}_t) - f(\mathbf{x}^*) &\leq \frac{1}{\sum_{k=0}^{t-1} \eta_k} \sum_{k=0}^{t-1} \eta_k (f(\mathbf{x}_k) - f(\mathbf{x}^*)) \\
&\leq \frac{1}{\sum_{t=0}^{T-1} \eta_t} \sum_{t=0}^{T-1} \eta_t \left( \langle \mathbf{x}_t - \mathbf{x}_*, \mathbf{g}_t \rangle + \langle \mathbf{x}_t - \mathbf{x}_*, \nabla f(\mathbf{x}_t) - \mathbf{g}_t \rangle \right) \\
&\leq \frac{1}{\sum_{t=0}^{T-1} \eta_t} \left\{ \underbrace{\sum_{k=0}^{t-1} [B_{\psi_k}(\mathbf{x}^*, \mathbf{x}_k) - B_{\psi_k}(\mathbf{x}^*, \mathbf{x}_{k+1})]}_{I_1} + \underbrace{\frac{1}{2} \sum_{k=0}^{t-1} \eta_k^2 \|\mathbf{g}_k\|_{\psi_k^*}^2}_{I_2} \right. \\
&\quad \left. + \underbrace{\sum_{k=0}^{t-1} \eta_k \langle \mathbf{x}_k - \mathbf{x}^*, \Delta_k \rangle}_{noise} \right\}, \quad\quad (B.3)
\end{aligned}
$$

where the first inequality holds by the convexity of $f(\mathbf{x})$ and Jensen's inequality, the second inequality follows again from the convexity of $f(\mathbf{x})$, and the last inequality follows from (B.2). For $I_1$ on the right-hand side of (B.3), denote $D_t = \max_{i \leq t} \|\mathbf{x}_i - \mathbf{x}^*\|_\infty$, we have

$$
\begin{aligned}
\sum_{k=0}^{t-1} B_{\psi_k}(\mathbf{x}^*, \mathbf{x}_k) - B_{\psi_k}(\mathbf{x}^*, \mathbf{x}_{k+1}) &= \sum_{i=1}^{d} \sum_{k=0}^{t-1} s_{k,i}(d_{k,i}^2 - d_{k+1,i}^2)/2 \\
&\leq D_t^2 \sum_{i=1}^{d} s_{t-1,i}, \quad\quad (B.4)
\end{aligned}
$$

where the equality holds due to triangle inequality.

For $I_2$ on the right-hand side of (B.3), we have

$$
\begin{aligned}
\sum_{k=0}^{t-1} \eta_k^2 \|\mathbf{g}_t\|_{\psi_k^*}^2 &\leq \eta_t^2 \sum_{i=1}^{d} \sum_{k=0}^{t-1} \frac{g_{k,i}^2}{s_{k,i}} \\
&\leq 2\eta_t^2 \sum_{i=1}^{d} s_{t-1,i} \\
&= O(D_t^2 \sum_{i=1}^{d} s_{t-1,i}), \quad\quad (B.5)
\end{aligned}
$$

where the first inequality holds due to the nondecreasing property of $\eta_t$, and the second inequality holds by using Lemma E.2 and the definition of $s_{k,i}$ for every $i = 1, \cdots, d$, and the equality follows

from the fact that

$$\eta_t \leq \max_{k \leq t} \|\mathbf{x}_t - \mathbf{x}_0\|_2 / \sqrt{d} + \epsilon \leq \max_{k \leq t} \|(\mathbf{x}_t - \mathbf{x}^*) - (\mathbf{x}_0 - \mathbf{x}^*)\|_2 / \sqrt{d} + \epsilon \leq D_t.$$

For the *noise* term of (B.3), let

$$Y_k = \eta_k \overline{D}_k, \ X_k = \left\langle \Delta_k, \frac{\mathbf{x}_k - \mathbf{x}_*}{\overline{D}_k} \right\rangle, \ \text{and} \ \widehat{X}_k = -\left\langle \nabla f(\mathbf{x}_k), \frac{\mathbf{x}_k - \mathbf{x}_*}{\overline{D}_k} \right\rangle,$$

then we get

$$\sum_{k=0}^{t-1} Y_k X_k = \sum_{k=0}^{t-1} \eta_k \langle \Delta_k, \mathbf{x}_k - \mathbf{x}_* \rangle.$$

Therefore, by Lemma E.3, we have

$$\mathbb{P}\left( \exists t \leq T : \left| \sum_{k=0}^{t-1} \eta_k \langle \Delta_k, \mathbf{x}_k - \mathbf{x}_* \rangle \right| \geq 8\eta_{t-1} \overline{D}_{t-1} \sqrt{\theta_{t,\delta} \sum_{i=1}^{d} s_{t-1,i}^2 + L^2 \theta_{t,\delta}^2} \right)$$

$$\leq \mathbb{P}\left( \exists t \leq T : \left| \sum_{k=0}^{t-1} Y_k X_k \right| \geq 8 Y_t \sqrt{\theta_{t,\delta} \sum_{k=0}^{t-1} (X_{k-1} - \widehat{X}_{k-1})^2 + L^2 \theta_{t,\delta}^2} \right)$$

$$\leq \delta + \mathbb{P}(\bar{l}_T \geq L), \tag{B.6}$$

where $\bar{l}_T = \max_{t \leq T} l(\mathbf{x}_t)$.

By substituting (B.5),(B.4) and (B.6) into (B.3), we have that, for all $\delta \in (0,1)$ and $L > 0$, with probability at least $1 - \delta - \mathbb{P}(\bar{l}_T > L)$, for all $t \leq T$, the optimality gap $f(\overline{\mathbf{x}}_t) - f_*$ is

$$O\left( \frac{D_t^2 \sum_{i=1}^{d} s_{t,i} / \eta_t + 8\overline{D}_t \sqrt{\theta_{t,\delta} \sum_{i=1}^{d} s_{t,i}^2 + L^2 \theta_{t,\delta}^2}}{\sum_{k=0}^{t-1} \eta_k / \eta_t} \right). \tag{B.7}$$

For the numerator, we use the QM-AM inequality to obtain the bound of $\sum_{i=1}^{d} s_{t,i} \leq \sqrt{d} \|\mathbf{s}_t\|_2$, and due to the non-decreasing property of $\eta_t$, we get

$$D_t^2 \sum_{i=1}^{d} s_{t,i} / \eta_t = D_t^2 \|\mathbf{s}_t\|_1 \leq \frac{D_t^2 \sqrt{d}}{\eta_0} \|\mathbf{s}_t\|_2, \tag{B.8}$$

where $\theta_{t,\delta} = \log(\frac{60 \log(6t)}{\delta})$. For the denominator, by applying Lemma E.1 for $\frac{\eta_t}{\sum_{k=0}^{t-1} \eta_k}$ and using the fact that $\eta_0 < \eta_t$, we attain that, for $\tau \in \arg\max_{t \leq T} \sum_{i=0}^{t-1} \frac{\eta_i}{\eta_t}$,

$$\frac{\eta_\tau}{\sum_{k=0}^{\tau-1} \eta_k} \leq \frac{1}{T} \log(\frac{\eta_T}{\eta_0}). \tag{B.9}$$

By substituting (B.8) and (B.9) into(B.7), we complete the proof. $\qquad \square$

## C    PROOF OF THEOREM 5.1

*Proof of Theorem 5.1.* For step $k$, we define $\mathbf{d}_k = \mathbf{x}_k - \mathbf{x}_*$, and let $\psi_k(\mathbf{x}) = \langle \mathbf{x}, \mathbf{H}_k \mathbf{x} \rangle$ and $B_\psi(\mathbf{x}, \mathbf{y}) = \psi(\mathbf{x} - \mathbf{y})/2$. Let $\mathbf{g}_k = (g_{k,1}, \cdots, g_{k,d}), \mathbf{s}_k = (s_{k,1}, \cdots, s_{k,d}), \mathbf{v}_k = (v_{k,1}, \cdots, v_{k,d})$ and $\mathbf{d}_k = (d_{k,1}, \cdots, d_{k,d})$. From the definition of $\mathbf{x}_{k+1}$, similar to (B.1), we have

$$\mathbf{x}_{k+1} = \arg\min_{\mathbf{x}} \{\eta_k \langle \mathbf{m}_k, \mathbf{x} \rangle + B_{\psi_k}(\mathbf{x}, \mathbf{x}_k)\},$$

which gives

$$\langle \mathbf{x} - \mathbf{x}_{k+1}, \eta_k \mathbf{m}_k + \nabla \psi_k(\mathbf{x}_{k+1}) - \nabla \psi_k(\mathbf{x}_k) \rangle \geq 0$$

for all $\mathbf{x}$. Setting $\mathbf{x} = \mathbf{x}^*$ and rearranging the terms, we can then obtain a bound of $\langle \mathbf{x}_{k+1} - \mathbf{x}^*, \mathbf{m}_t \rangle$. Thus similar to (B.2), we have the inequality by denoting the dual norm of $\|\cdot\|_{\psi_k}$ by $\|\cdot\|_{\psi_k^*}$

$$\eta_k \langle \mathbf{x}_k - \mathbf{x}^*, \mathbf{m}_k \rangle \leq B_{\psi_k}(\mathbf{x}^*, \mathbf{x}_k) - B_{\psi_k}(\mathbf{x}^*, \mathbf{x}_{k+1}) + \frac{\eta_k^2}{2} \|\mathbf{m}_k\|_{\psi_k^*}^2.$$

Using the fact that $\mathbf{m}_k = \beta_{1k} \mathbf{m}_{k-1} + (1 - \beta_{1k}) \mathbf{g}_k$, we have

$$\eta_k \langle \mathbf{x}_k - \mathbf{x}^*, \mathbf{g}_k \rangle \leq \frac{1}{1 - \beta_{1k}} (B_{\psi_k}(\mathbf{x}^*, \mathbf{x}_k) - B_{\psi_k}(\mathbf{x}^*, \mathbf{x}_{k+1}))$$

$$+ \frac{\eta_k^2}{2(1 - \beta_1)} \|\mathbf{m}_k\|_{\psi_k^*}^2 - \frac{\eta_k \beta_{1k}}{1 - \beta_{1k}} \langle \mathbf{x}_k - \mathbf{x}^*, \mathbf{m}_{k-1} \rangle$$

$$\leq \frac{1}{1 - \beta_{1k}} (B_{\psi_k}(\mathbf{x}^*, \mathbf{x}_k) - B_{\psi_k}(\mathbf{x}^*, \mathbf{x}_{k+1})) + \frac{\eta_k^2}{2(1 - \beta_{1k})} \|\mathbf{m}_k\|_{\psi_k^*}^2$$

$$+ \frac{\eta_k^2 \beta_{1k}}{2(1 - \beta_{1k})} \|\mathbf{m}_{k-1}\|_{\psi_k^*}^2 + \frac{\beta_{1k}}{1 - \beta_{1k}} B_{\psi_k}(\mathbf{x}_k, \mathbf{x}^*). \tag{C.1}$$

By the convexity of $f(\mathbf{x})$, we have

$$f(\overline{\mathbf{x}}_t) - f(\mathbf{x}^*) \leq \frac{1}{\sum_{k=0}^{t-1} \eta_k} \sum_{k=0}^{t-1} \eta_k \langle \mathbf{x}_k - \mathbf{x}^*, \nabla f(\mathbf{x}_k) \rangle$$

$$= \frac{1}{\sum_{k=0}^{t-1} \eta_k} (\sum_{k=0}^{t-1} \eta_k \langle \mathbf{x}_k - \mathbf{x}^*, \mathbf{g}_k \rangle + \eta_k \langle \mathbf{x}_k - \mathbf{x}^*, \Delta_k \rangle),$$

where $\Delta_t = \nabla f(\mathbf{x}_t) - \mathbf{g}_t$. Substituting (C.1) into the above inequality leads to

$$f(\overline{\mathbf{x}}_t) - f(\mathbf{x}^*) \leq \frac{1}{\sum_{k=0}^{t-1} \eta_k} \Bigg\{ \underbrace{\sum_{k=0}^{t-1} \frac{(B_{\psi_k}(\mathbf{x}^*, \mathbf{x}_k) - B_{\psi_k}(\mathbf{x}^*, \mathbf{x}_{k+1}))}{(1 - \beta_{1k})}}_{I_1} + \underbrace{\sum_{k=0}^{t-1} \frac{\beta_{1k}}{1 - \beta_{1k}} B_{\psi_k}(\mathbf{x}_k, \mathbf{x}^*)}_{I_2}$$

$$+ \underbrace{\sum_{k=0}^{t-1} (\frac{\eta_k^2}{2(1 - \beta_{1k})} \|\mathbf{m}_k\|_{\psi_k^*}^2 + \frac{\eta_k^2 \beta_{1k}}{2(1 - \beta_{1k})} \|\mathbf{m}_{k-1}\|_{\psi_k^*}^2)}_{I_3} + \underbrace{\sum_{k=0}^{t-1} \eta_k \langle \mathbf{x}_k - \mathbf{x}^*, \Delta_k \rangle}_{\text{noise}} \Bigg\}. \tag{C.2}$$

For $I_1$, define $D_t = \max_{i \leq t} \|\mathbf{x}_i - \mathbf{x}^*\|_\infty$, and then we have

$$\sum_{k=0}^{t-1} \frac{B_{\psi_k}(\mathbf{x}^*, \mathbf{x}_k) - B_{\psi_k}(\mathbf{x}^*, \mathbf{x}_{k+1})}{1 - \beta_{1k}} \leq \sum_{i=1}^{d} \sum_{k=0}^{t-1} \frac{s_{k,i}(d_{k,i}^2 - d_{k+1,i}^2)}{2(1 - \beta_1)}$$

$$\leq \sum_{i=1}^{d} \sum_{k=0}^{t-1} \frac{s_{k,i} D_t^2}{1 - \beta_1}, \tag{C.3}$$

where the first inequality holds for the reason that $\beta_{1k} \leq \beta_1$, the second inequality holds for the definition of $D_t$ and the fact that $D_t > d_{k,i}$ for all $k < t$.

For $I_2$, by the definition of $B_{\psi_k}$, we have

$$B_{\psi_k}(\mathbf{x}_k, \mathbf{x}^*) \leq \frac{D_k^2}{2} \sum_{i=1}^{d} s_{k,i}.$$

And by the fact of $\beta_1 k = \beta_1 \lambda^k$, we have

$$\sum_{k=0}^{t-1} \frac{\beta_{1k}}{1 - \beta_{1k}} B_{\psi_k}(\mathbf{x}_k, \mathbf{x}^*) \leq \frac{\beta_1 D_t^2}{2(1 - \beta_1)(1 - \lambda)} \sum_{i=1}^{d} s_{t-1,i}. \tag{C.4}$$

For $I_3$ in the inequality (C.2), we give the proofs for the two cases in Algorithm **3** separately.

**Case 1:** $\mathbf{s}_t = (\sum_{k=0}^t \mathbf{g}_k^2)^{1/2}$.
If we choose the first definition of $\mathbf{s}_t$, we have the fact that

$$
\begin{aligned}
\|\mathbf{m}_t\|_{\psi_t^*}^2 &= \sum_{i=1}^d \frac{(\sum_{j=0}^t (1-\beta_{1j})\Pi_{s=1}^{t-j}\beta_{1(t-s+1)}g_{j,i})^2}{\sqrt{\sum_{j=0}^t g_{j,i}^2}} \\
&\leq \sum_{i=1}^d \frac{(\sum_{j=0}^t \Pi_{s=1}^{t-j}\beta_{1(t-s+1)})(\sum_{j=0}^t \Pi_{s=1}^{t-j}\beta_{1(t-s+1)}g_{j,i}^2)}{\sqrt{\sum_{j=0}^t g_{j,i}^2}} \\
&\leq \sum_{i=1}^d \frac{(\sum_{j=0}^t \beta_1^{t-j})(\sum_{j=0}^t \beta_1^{t-j}g_{j,i}^2)}{\sqrt{\sum_{j=0}^t g_{j,i}^2}} \\
&\leq \frac{1}{1-\beta_1}\sum_{i=1}^d \frac{\sum_{j=0}^t \beta_1^{t-j}g_{j,i}^2}{\sqrt{\sum_{j=0}^t g_{j,i}^2}},
\end{aligned}
\tag{C.5}
$$

where the first inequality follows from Cauchy-Schwarz inequality; the second inequality is due to the fact that $\beta_{1j} \leq \beta_1$ for all $j \leq t$; and the third inequality follows from the inequality $\sum_{j=1}^t \beta_1^{t-j} \leq 1/(1-\beta_1)$.

By summing the inequalities in (C.5) from $k=0$ to $k=t-1$, we obtain

$$
\begin{aligned}
\sum_{k=0}^{t-1} \|\mathbf{m}_k\|_{\psi_k^*}^2 &\leq \frac{1}{1-\beta_1}\sum_{i=1}^d \sum_{k=0}^{t-1} \frac{\sum_{j=0}^k \beta_1^{k-j}g_{j,i}^2}{\sqrt{\sum_{j=0}^k g_{j,i}^2}} \\
&\leq \frac{1}{1-\beta_1}\sum_{i=1}^d \underbrace{\sum_{k=0}^{t-1}\sum_{j=0}^k \frac{\beta_1^{k-j}g_{j,i}^2}{\sqrt{\sum_{s=0}^j g_{s,i}^2}}}_{I_i},
\end{aligned}
$$

where the second inequality holds by the fact that $\sqrt{\sum_{j=0}^k g_{j,i}^2} \geq \sqrt{\sum_{s=0}^j g_{s,i}^2}$ for $j \leq k$.

$$
\begin{aligned}
\sum_{k=0}^{t-1}\sum_{j=1}^k \frac{\beta_1^{k-j}g_{j,i}^2}{\sqrt{\sum_{s=0}^j g_{s,i}^2}} &= \sum_{j=0}^{t-1}\sum_{k=j}^{t-1} \frac{\beta_1^{k-j}g_{j,i}^2}{\sqrt{\sum_{s=0}^j g_{s,i}^2}} \\
&= \sum_{j=0}^{t-1}\sum_{k=0}^{t-1-j} \frac{\beta_1^{k}g_{j,i}^2}{\sqrt{\sum_{s=0}^j g_{s,i}^2}} \\
&= \sum_{j=0}^{t-1}(\sum_{k=0}^{t-1-j} \beta_1^{k})\frac{g_{j,i}^2}{\sqrt{\sum_{s=0}^j g_{s,i}^2}},
\end{aligned}
$$

where the first equality holds by re-arrangement and the second equality holds by replace $k$ with $k-j$. Noting that $\frac{g_{j,i}^2}{\sqrt{\sum_{k=0}^j g_{j,i}^2}} \leq 2(\sqrt{\sum_{k=0}^j g_{j,i}^2} - \sqrt{\sum_{k=0}^{j-1} g_{j,i}^2})$, then we have

$$
\begin{aligned}
\sum_{k=0}^{t-1} \|\mathbf{m}_k\|_{\psi_j^*}^2 &\leq \frac{1}{1-\beta_1}\sum_{i=1}^d \sum_{j=0}^{t-1}(\sum_{k=0}^{t-1-j} \beta_1^{k})\frac{g_{j,i}^2}{\sqrt{\sum_{s=0}^j g_{s,i}^2}} \\
&\leq \frac{2}{1-\beta_1}\sum_{i=1}^d \sum_{j=0}^{t-1} \beta_1^{t-1-j}\sqrt{\sum_{s=0}^j g_{s,i}^2}. \\
&\leq \frac{2}{(1-\beta_1)^2}\|\mathbf{s}_{t-1}\|_1,
\end{aligned}
\tag{C.6}
$$

where the last inequality follows the facts of $\sqrt{\sum_{s=0}^{j} g_{s,i}^2} = \|\mathbf{s}_j\|_1$ and the non-decreasing property of $\|\mathbf{s}_t\|_1$.

**Case 2:** $\mathbf{s}_t = \sqrt{(t+1) \cdot \max_{k \leq t}(\mathbf{v}_k)}$.

If we choose the second form of $\mathbf{s}_t$, we have

$$\|\mathbf{m}_k\|_{\psi_k^*}^2 = \sum_{i=1}^{d} \frac{m_{k,i}^2}{s_{k,i}}$$

$$\leq \sum_{i=1}^{d} \frac{m_{k,i}^2}{\sqrt{(k+1)v_{k,i}}}$$

$$= \sum_{i=1}^{d} \frac{(\sum_{j=0}^{k}(1-\beta_{1j})\Pi_{s=1}^{k-j}\beta_{1(k-s+1)}g_{j,i})^2}{\sqrt{(k+1)((1-\beta_2)\sum_{j=0}^{k}\beta_2^{k-j}g_{j,i}^2)}},$$

where the first equality and inequality follow by the definitions of $\mathbf{m}_k$ and $s_{k,i}$, and the second equality follow by the update rule of $m_{k,i}$ and $v_{k,i}$ for Case 2.

For the definition of $\mathbf{s}_t$, by the Cauchy-Schwarz inequality and the fact that $\beta_{1t} \leq \beta_1$, we have

$$\|\mathbf{m}_k\|_{\psi_k^*}^2 \leq \sum_{i=1}^{d} \frac{(\sum_{j=0}^{k} \Pi_{s=1}^{k-j}\beta_{1(k-s+1)})(\sum_{j=0}^{k} \Pi_{s=1}^{k-j}\beta_{1(k-s+1)}g_{j,i}^2)}{\sqrt{(k+1)((1-\beta_2)\sum_{j=0}^{k}\beta_2^{k-j}g_{j,i}^2)}}$$

$$\leq \sum_{i=1}^{d} \frac{(\sum_{j=0}^{k}\beta_1^{k-j})(\sum_{j=0}^{k}\beta_1^{k-j}g_{j,i}^2)}{\sqrt{(k+1)((1-\beta_2)\sum_{j=0}^{k}\beta_2^{k-j}g_{j,i}^2)}}.$$

Then by applying the inequality that $\sum_{j=0}^{k}\beta_1^{k-j} \leq 1/(1-\beta_1)$ and denoting $\gamma = \beta_1/\sqrt{\beta_2} < 1$, we have

$$\|\mathbf{m}_k\|_{\psi_k^*}^2 \leq \frac{1}{(1-\beta_1)\sqrt{(k+1)(1-\beta_2)}} \sum_{i=1}^{d} \frac{\sum_{j=0}^{k}\beta_1^{k-j}g_{j,i}^2}{\sqrt{\sum_{j=0}^{k}\beta_2^{k-j}g_{j,i}^2}}$$

$$\leq \frac{1}{(1-\beta_1)\sqrt{(k+1)(1-\beta_2)}} \sum_{i=1}^{d} \sum_{j=0}^{k} \frac{\beta_1^{k-j}g_{j,i}^2}{\sqrt{\beta_2^{k-j}g_{j,i}^2}}$$

$$\leq \frac{1}{(1-\beta_1)\sqrt{(k+1)(1-\beta_2)}} \sum_{i=1}^{d} \sum_{j=0}^{k} \gamma^{k-j}|g_{j,i}|.$$

Thus by summing up $\|\mathbf{m}_t\|_{\psi_t^*}^2$, we have:

$$\sum_{k=0}^{t} \|\mathbf{m}_k\|_{\psi_k^*}^2 \leq \sum_{k=0}^{t} \frac{1}{(1-\beta_1)\sqrt{(k+1)(1-\beta_2)}} \sum_{i=1}^{d} \sum_{j=0}^{k} \gamma^{k-j}|g_{j,i}|$$

$$= \frac{1}{(1-\beta_1)\sqrt{1-\beta_2}} \sum_{i=1}^{d} \sum_{k=0}^{t} |g_{k,i}| \sum_{j=k}^{t} \frac{\gamma^{j-k}}{\sqrt{j+1}}$$

$$\leq \frac{1}{(1-\beta_1)\sqrt{1-\beta_2}} \sum_{i=1}^{d} \sum_{k=0}^{t} |g_{k,i}| \sum_{j=k}^{t} \frac{\gamma^{j-k}}{\sqrt{k+1}}$$

$$\leq \frac{1}{(1-\beta_1)\sqrt{1-\beta_2}} \sum_{k=0}^{t} \frac{\|\mathbf{g}_k\|_1}{(1-\gamma)\sqrt{(k+1)}}, \tag{C.7}$$

where the equality holds by re-arranging the terms. the second inequality holds by the fact that $\sqrt{j+1} \geq \sqrt{k+1}$ when $j \geq k$, and the last inequality holds by the definition of 1-norm and

the fact that $\sum_{j=k}^{t} \gamma^{j-k} \leq \sum_{j=0}^{\infty} \gamma^j = \frac{1}{1-\gamma}$ when $0 < \gamma < 1$. For the noise term, we define $\overline{D}_t = \max k \leq t \|\mathbf{d}_k\|_2$, and let

$$Y_k = \eta_k \overline{D}_k, \ X_k = \left\langle \Delta_k, \frac{\mathbf{x}_k - \mathbf{x}_*}{\overline{D}_k} \right\rangle, \text{ and } \widehat{X}_k = -\left\langle \nabla f(\mathbf{x}_k), \frac{\mathbf{x}_k - \mathbf{x}_*}{\overline{D}_k} \right\rangle.$$

Thus we get

$$\sum_{k=0}^{t-1} Y_k X_k = \sum_{k=0}^{t-1} \eta_k \langle \Delta_k, \mathbf{x}_k - \mathbf{x}_* \rangle.$$

Therefore, we have

$$\mathbb{P}\left( \exists t \leq T : \left| \sum_{k=0}^{t-1} \eta_k \langle \Delta_k, \mathbf{x}_k - \mathbf{x}_* \rangle \right| \geq 8\eta_{t-1} \overline{D}_{t-1} \sqrt{\theta_{t,\delta} \sum_{i=1}^{d} s_{t,i}^2 + L^2 \theta_{t,\delta}^2} \right)$$

$$\leq \mathbb{P}\left( \exists t \leq T : \left| \sum_{k=0}^{t-1} Y_k X_k \right| \geq 8Y_t \sqrt{\theta_{t,\delta} \sum_{k=0}^{t-1} (X_k - \widehat{X}_k)^2 + L^2 \theta_{t,\delta}^2} \right)$$

$$\leq \delta + \mathbb{P}(\bar{l}_T \geq L),$$

where the last inequality holds by using lemma E.3 and defining $\bar{l}_t = \max_{k \leq t} l(\mathbf{x}_k)$. Thus we have that, for all $\delta \in (0,1)$ and $L > 0$, with probability at least $1 - \delta - \mathbb{P}(\bar{l}_T > L)$, for all $t \leq T$,

$$f(\overline{\mathbf{x}}_t) - f_* \leq (I_1 + I_2 + I_3) + \frac{8\overline{D}_t \eta_t}{\sum_{k=0}^{t-1} \eta_k} \sqrt{\theta_{t,\delta} \sum_{i=1}^{d} s_{t,i}^2 + L^2 \theta_{t,\delta}^2}. \tag{C.8}$$

Thus by substituting (C.3),(C.4), (C.6) and (C.7) into (C.8), we obtain that for all $t < T$

$$f(\overline{\mathbf{x}}_t) - f_* \leq \frac{\eta_t}{\sum_{k=0}^{t-1} \eta_k} \left( \frac{D_t^2}{(1-\beta_1)\eta_t} \|\mathbf{s}_{t-1}\|_1 + \frac{(1+\beta_1)\eta_t}{(1-\beta_1)^3} \|\mathbf{s}_{t-1}\|_1 \right.$$

$$\left. + \frac{\beta_1 D_t^2 \|\mathbf{s}_{t-1}\|_1}{2(1-\beta_1)(1-\lambda)\eta_t} + 8\overline{D}_t \sqrt{\theta_{t,\delta} \|\mathbf{s}_t\|_2^2 + L^2 \theta_{t,\delta}^2} \right)$$

for case 1, and

$$f(\overline{\mathbf{x}}_t) - f_* \leq \frac{\eta_t}{\sum_{k=0}^{t-1} \eta_k} \left( \frac{D_t^2}{(1-\beta_1)\eta_t} \|\mathbf{s}_{t-1}\|_1 + \frac{(1+\beta_1)\eta_t}{(1-\beta_1)^2 \sqrt{1-\beta_2}(1-\gamma)} \sum_{k=0}^{t-1} \frac{\|\mathbf{g}_k\|_1}{\sqrt{k+1}} \right.$$

$$\left. + \frac{\beta_1 D_t^2 \|\mathbf{s}_{t-1}\|_1}{2(1-\beta_1)(1-\lambda)\eta_t} + 8\overline{D}_t \sqrt{\theta_{t,\delta} \|\mathbf{s}_t\|_2^2 + L^2 \theta_{t,\delta}^2} \right)$$

for case 2, with probability at least $1 - \delta - \mathbb{P}(\bar{l}_T > L)$. By (B.8) and (B.9), for $\tau \in \arg\max_{t \leq T} \sum_{i=0}^{t-1} \frac{\eta_i}{\eta_t}$, we have,

$$f(\overline{\mathbf{x}}_\tau) - f^* \leq O\left( \log\left( \frac{\eta_T}{\eta_0} \right) \left( \frac{D_\tau^2 \sqrt{d} \|\mathbf{s}_\tau\|_2}{(1-\beta_1)\eta_0} + \frac{(1+\beta_1)D_\tau \sqrt{d}}{(1-\beta_1)^3} \|\mathbf{s}_\tau\|_2 \right.\right.$$

$$\left.\left. + \frac{\beta_1 D_\tau^2 \sqrt{d} \|\mathbf{s}_\tau\|_2}{2(1-\beta_1)(1-\lambda)\eta_0} + 8\overline{D}_\tau \sqrt{\theta_{\tau,\delta} \|\mathbf{s}_\tau\|_2^2 + L^2 \theta_{\tau,\delta}^2} \right)/T \right)$$

for case 1 and

$$f(\overline{\mathbf{x}}_\tau) - f^* \leq O\left( \log\left( \frac{\eta_T}{\eta_0} \right) \left( \frac{D_\tau^2 \sqrt{d} \|\mathbf{s}_\tau\|_2}{(1-\beta_1)\eta_0} + \frac{(1+\beta_1)D_\tau \sqrt{d}}{(1-\beta_1)^2 \sqrt{1-\beta_2}(1-\gamma)} \sum_{k=0}^{\tau-1} \frac{\|\mathbf{g}_k\|_2}{\sqrt{k+1}} \right.\right.$$

$$\left.\left. + \frac{\beta_1 D_\tau^2 \sqrt{d} \|\mathbf{s}_\tau\|_2}{2(1-\beta_1)(1-\lambda)\eta_0} + 8\overline{D}_\tau \sqrt{\theta_{\tau,\delta} \|\mathbf{s}_\tau\|_2^2 + L^2 \theta_{\tau,\delta}^2} \right)/T \right)$$

for case 2, with probability at least $1 - \delta - \mathbb{P}(\bar{l}_T > L)$. Then we finish the proof.

$$\square$$

# D CONVERGENCE ANALYSIS FOR NONCONVEX OPTIMIZATION

In this section, we provide convergence guarantees for AdaGrad++ and Adam++ in the nonconvex deterministic optimization setting.

## D.1 PRELIMINARIES

For the parameter $\mathbf{x} \in \mathbb{R}^d$, we denote $\mathbf{x}_* = \arg\min_{\mathbf{x}} f(\mathbf{x}) \geq -\infty$. And in this section, $\|\cdot\|$ denotes the $\ell_2$ norm by default. Furthermore, we assume the following assumptions hold for the proposed optimizers:

**Assumption D.1.** The gradient of $f(\mathbf{x})$ is bounded by a constant $G > 0$, i.e., $\|\nabla f(\mathbf{x})\| \leq G$.

This assumption can be potentially relaxed to Assumption 4.1, which we leave it as a future work.

**Assumption D.2.** The function $f(\mathbf{x})$ is $L$-smooth, i.e.,

$$f(\mathbf{x}) \leq f(\mathbf{y}) + \langle \nabla f(\mathbf{y}), \mathbf{x} - \mathbf{y} \rangle + \frac{L}{2}\|\mathbf{x} - \mathbf{y}\|^2, \forall \mathbf{x}, \mathbf{y} \in \mathbb{R}^d.$$

Note that if a function $f(\mathbf{x})$ is $L$-smooth, we have for any $\mathbf{x}, \mathbf{y} \in \mathbb{R}^d$, $\|\nabla f(\mathbf{x}) - \nabla f(\mathbf{y})\| \leq L\|\mathbf{x} - \mathbf{y}\|_2$.

## D.2 CONVERGENCE RESULTS

For nonconvex optimization setting of AdaGrad++, we have the following theorem:

**Theorem D.3** (Convergence of AdaGrad++). Let $\mathbf{x}_0, \cdots, \mathbf{x}_T$ be the iterates of AdaGrad++, and $\mathbf{g}_0, \cdots, \mathbf{g}_T$ be the corresponding gradients. Denote $f_k = f(\mathbf{x}_k), f_* = f(\mathbf{x}_*)$. Under Assumptions D.1 and D.2, for any $t \geq 1$, it holds that

$$\min_{0 \leq k \leq t} \|\mathbf{g}_k\|_1 \leq \frac{\frac{f_0 - f_*}{\eta_0} + \widetilde{D}_t \sum_{i=1}^d (2\log(\frac{G\sqrt{t}}{\mathbf{g}_{0,i}}) + 1)}{\sqrt{t}},$$

where $\widetilde{D}_t = \max_{\tau \leq t} \|\mathbf{x}_t - \mathbf{x}_0\|/\sqrt{d}$.

*Proof of Theorem D.3.* By Assumption D.2, for all $k \geq 0$ we have

$$\frac{f_{k+1} - f_k}{\eta_k} \leq -\langle \nabla f(\mathbf{x}_k), \mathbf{H}_k^{-1}\mathbf{g}_k \rangle + \frac{\eta_k L}{2}\|\mathbf{H}_k^{-1}\mathbf{g}_k\|^2.$$

Thus by rearranging, and by the definition of $\mathbf{g}_k$, we have

$$\langle \mathbf{g}_k, \mathbf{H}_k^{-1}\mathbf{g}_k \rangle \leq \frac{f_k - f_{k+1}}{\eta_k} + \frac{\eta_k L}{2}\|\mathbf{H}_k^{-1}\mathbf{g}_k\|^2. \tag{D.1}$$

Take the sum of (D.1) from $k = 0$ to $t - 1$, and by the definition of $\mathbf{H}_k$, we have

$$\sum_{k=0}^{t-1}\sum_{i=1}^d \frac{\mathbf{g}_{k,i}^2}{\mathbf{s}_{k,i}} \leq \underbrace{\sum_{k=0}^{t-1} \frac{f_k - f_{k+1}}{\eta_k}}_{I_1} + \frac{L}{2}\underbrace{\sum_{k=0}^{t-1} \eta_k \|\mathbf{H}_k^{-1}\mathbf{g}_k\|^2}_{I_2}. \tag{D.2}$$

For the $I_1$ on the right-hand side of (D.2), by the facts that $\eta_k$ is nondecreasing and $f_*$ is the global minimal value, we have

$$\sum_{k=0}^{t-1} \frac{f_k - f_{k+1}}{\eta_k} = \frac{f_0}{\eta_0} - \sum_{k=1}^{t-1} f_k \cdot \left(\frac{1}{\eta_{k-1}} - \frac{1}{\eta_k}\right) - \frac{f_t}{\eta_{t-1}}$$

$$\leq \frac{f_0}{\eta_0} - \sum_{k=1}^{t-1} f_* \cdot \left(\frac{1}{\eta_{k-1}} - \frac{1}{\eta_k}\right) - \frac{f_*}{\eta_{t-1}}$$

$$= \frac{f_0 - f_*}{\eta_0}. \tag{D.3}$$

For the $I_2$ in (D.2), we have

$$\sum_{k=0}^{t-1} \eta_k \|\mathbf{H}_k^{-1}\mathbf{g}_k\|^2 = \sum_{k=0}^{t-1} \eta_k \sum_{i=1}^{d} \frac{\mathbf{g}_{k,i}^2}{\mathbf{s}_{k,i}^2}$$

$$\leq \eta_{t-1} \sum_{i=1}^{d} \sum_{k=0}^{t-1} \frac{\mathbf{g}_{k,i}^2}{\mathbf{s}_{k,i}^2}$$

$$\leq \eta_{t-1} \sum_{i=1}^{d} \left[ 2\log\left(\frac{\mathbf{s}_{t,i}}{\mathbf{g}_{0,i}}\right) + 1 \right]$$

$$\leq \eta_{t-1} \sum_{i=1}^{d} \left[ 2\log\left(\frac{\sqrt{t}\cdot G}{\mathbf{g}_{0,i}}\right) + 1 \right], \tag{D.4}$$

where the first inequality holds due to the fact that $\eta_k$ is non-decreasing, the second inequality holds by using Lemma E.2 with $a_l = g_{l,i}^2/g_{0,i}^2$, and the third inequality holds follows from the fact that $\mathbf{s}_{t,i} \leq \sqrt{t}\cdot G$ by the definition of $\mathbf{s}_t$ and Assumption D.1. For the term on the left-hand side of (D.2), we have

$$\sum_{k=0}^{t-1}\sum_{i=1}^{d} \frac{\mathbf{g}_{k,i}^2}{\mathbf{s}_{k,i}} \geq \sum_{i=1}^{d}\sum_{k=0}^{t-1} \frac{\mathbf{g}_{k,i}^2}{\mathbf{s}_{t,i}} = \sum_{i=1}^{d} \mathbf{s}_{t,i} \geq \sum_{i=1}^{d}\sum_{k=0}^{t-1} \frac{|\mathbf{g}_{k,i}|}{\sqrt{t}} \geq \sqrt{t}\min_{k<t}\|\mathbf{g}_k\|_1 \tag{D.5}$$

where the first inequality holds by the fact that $s_{k,i}$ is nondecreasing, the second inequality holds by the QM–AM inequality, and the last inequality holds by the definition of $\ell_1$ norm. Note that $\eta_t = \max(\eta_{t-1}, \|\mathbf{x}_t - \mathbf{x}_0\|_2/\sqrt{d})$, we thus complete the proof by substituting inequalities (D.3), (D.4), (D.5) into (D.2). $\square$

For nonconvex optimization, we have a similar convergence guarantee for Adam++:

**Theorem D.4** (Convergence of Adam++). Let $\mathbf{x}_0,\cdots,\mathbf{x}_T$ be the iterates of Adam++, and $\mathbf{g}_0,\cdots,\mathbf{g}_T$ be the corresponding gradients. Denote $f_k = f(\mathbf{x}_k), f_* = f(\mathbf{x}_*)$. Under Assumptions D.1 and D.2, for any $t \geq 1$, it holds that

$$\min_{0\leq k\leq t}\|\mathbf{g}_k\|_1 \leq \frac{\frac{f_0-f_*}{\eta_0} + \widetilde{D}_t \sum_{i=1}^{d}(2\log(\frac{G\sqrt{t}}{\mathbf{g}_{0,i}})+1)}{\sqrt{t}},$$

where $\widetilde{D}_t = \max_{\tau\leq t}\|\mathbf{x}_t - \mathbf{x}_0\|/\sqrt{d}$.

The proof of Theorem D.4 is similar to that of Theorem D.3, hence we omit it.

# E AUXILIARY LEMMAS

In this section, we present and summarize two auxiliary lemmas provided by Ivgi et al. (2023) that provide tools for our proof of the main theorems.

**Lemma E.1.** [Lemma 3 in Ivgi et al. (2023)] Suppose $0 < a_0 \leq a_1 \leq \cdots \leq a_T$. Then the following two inequalities hold:

$$\max_{t\leq T}\sum_{\tau<t} \frac{a_\tau}{a_t} \geq \frac{1}{e}\left(\frac{T}{\log_+(a_T/a_0)} - 1\right)$$

**Lemma E.2.** [Lemma 4 in Ivgi et al. (2023)] Suppose $0 < a_0 \leq a_1 \leq \cdots \leq a_T$. Then the following two inequalities hold:

$$\sum_{k=1}^{t} \frac{a_k - a_{k-1}}{\sqrt{a_k}} \leq 2(\sqrt{a_t} - \sqrt{a_0}).$$

**Lemma E.3.** [Lemma 7 in Ivgi et al. (2023)] Consider a filtration $\mathcal{F} = \{\mathcal{F}_t\}_{t\geq 0}$ in a probability space. Let $S$ be the set of nonnegative and nondecreasing sequences. Suppose that $C_t \in \mathcal{F}_{t-1}$ and that $\{X_t\}_{t\geq 0}$ is a martingale difference sequence adapted to $\{\mathcal{F}_t\}_{t\geq 0}$ such that $|X_t| \leq C_t$ with

probability 1 for all $t \geq 0$. Then, for all $\delta \in (0,1)$, $c > 0$, $T > 0$, and $\overline{X}_t \in \mathcal{F}_{t-1}$ such that $|\overline{X}_t| \leq C_t$ with probability 1, it holds that

$$\mathbb{P}\left(\exists t \leq T, \exists\{y_i\}_{i=1}^{\infty} \in S \text{ such that } \left|\sum_{i=1}^{t} y_i X_i\right| \geq 8y_t\sqrt{\theta_{t,\delta}\sum_{i=1}^{t}(X_i - \overline{X}_i)^2 + c^2\theta_{t,\delta}^2}\right)$$

$$\leq \delta + \mathbb{P}(\exists t \leq T : C_t \geq c),$$

where $\theta_{t,\delta} = \log(\frac{60\log(6t)}{\delta})$.

## F DISCUSSION ON LEARNING RATE SEARCH ON PARAMETER-FREE ALGORITHMS

In our computer vision experiments, we implement learning rate search for the base learning rate within a small range, instead of setting learning rate to 1.0 as suggested. Actually, most of the parameter-free algorithms have to tune some hyper-parameters. For example, DoG (Ivgi et al., 2023) may set different $r_\epsilon$s and D-Adapt Adam (Defazio and Mishchenko, 2023) can change different $\gamma_k$s.

However, as we shown in Appendix H.2, different hyper-parameters affect little on performance (but can still have some affect). For example, the test accuracy for Resnet18 model trained on CIFAR-100 with Adam++ (Case 1) only varies between 66.49 to 70.42 when the learning rate varies between 1.0 to 2.0. And the final learning rates are different for different algorithms and tasks. In comparison, for different scenarios, non-parameter-free algorithms, such as AdamW, requires different learning rates. For example, the learning rate for AdamW may vary from $\sim 10^{-2}$ in computer vision tasks to $\sim 10^{-4}$ in LLM tasks, since AdamW does not compute the optimal step size automatically.

The reason why we choose [0.1, 3.0] as the learning rate range is that we observed an obvious peak phenomenon during searching, and all the peaks in our experiments are located in [0.9, 2.0]. Therefore, we extend this range to [0.1, 3.0] for Adam++ and baseline parameter-free algorithms for fair comparison.

## G PARAMETER SETTINGS

Although the suggested learning rate is 1.0 for these parameter-free optimizers, throughout the training, we still search for the base learning rate from 0.1 to 3.0, and the initial learning rate of Adam++ is set to $1 \times 10^{-6}(1 + \|x_0\|_2^2)$ for image classification tasks, as suggested by Ivgi et al. (2023). For GPT-2 small and GPT-2 medium tasks, the initial learning rates of AdamW++ are $6 \times 10^{-4}(1 + \|x_0\|_2^2)$ and $3 \times 10^{-4}(1 + \|x_0\|_2^2)$, respectively, where $6 \times 10^{-4}$ and $3 \times 10^{-4}$ correspond to the default learning rates for AdamW training. The initial learning rates of Prodigy and D-Adapt Adam are set as the default $1 \times 10^{-6}$ as the algorithms did not suggest any modification of this parameter.

In addition, we list the parameters, architectures and hardware that we used for the experiments. All other parameters not listed are set as default. The information is collected in Tables 2–3.

Table 2: Computer Vision experiment.

| Hyper-parameter | Value |
| --- | --- |
| Architecture | ResNet 18, VGG16, DenseNet-121 |
| Epochs | 200 |
| GPUs | 1×A6000 |
| Batch size | 256 |
| LR schedule | Constant/Cosine Decay |
| weight decay | 5e-4 |
| $(\beta_1, \beta_2)$ | (0.9, 0.999) |

Table 3: Large language model experiment

| Hyper-parameter | Value |
| --- | --- |
| Architecture | GPT-2 Small/GPT-2 Medium |
| Steps | 50K |
| GPUs | 8×A100 |
| Batch size | 480 |
| Context Length | 1024 |
| LR schedule | Cosine Decay with Warmup |
| Seeds | 5000+offset |
| weight decay | 0.1 |
| $(\beta_1, \beta_2)$ | (0.9, 0.95) |
| Adam LR | 6e-4/3e-4 |

## H ADDITIONAL EXPERIMENTS

In this section, we present some additional experiment results.

## H.1 Results on More Network Models and Datasets

In this section, we include more experiments for additional network architectures, scheduler and datasets. Here we present (1) ResNet-18 model trained with CIFAR-10 dataset with Multi-step learning rate scheduler, which is similar to constant learning rate scheduler except for a $0.1\times$ learning rate decay at 100-th and 150-th steps (shown in Figures 11 and 12); (2) ResNet-18 and DenseNet-121 models trained with CIFAR-100 dataset with constant and cosine scheduler (presented in Figures 13 and 14); (3) ResNet-18 model trained with SVHN dataset (Netzer et al., 2011) with constant and cosine scheduler (displayed in Figures 15 and 16); and (4) ResNet-50 model (He et al., 2016) trained with Tiny-ImageNet dataset Tavanaei (2020) with constant and cosine scheduler (demonstrated in Figures 17 and 18). All other hyperparameters remain consistent with those detailed in Section 6.1 and Appendix G.

In Figures 11 and 12, we note that even with a more complicated scheduler, both Case 1 and Case 2 of Adam++ demonstrate a strong and consistent performance. And from Figures 13, 14, 15, 16, 17 and 18, it can be observed that Adam++ still outperforms other parameter-free algorithms in different kinds of computer vision tasks, including CIFAR-100, SVHN and Tiny-ImageNet.

This consistency in performance highlights Adam++'s robustness and its ability to perform reliably without the need for frequent adjustments or tuning.

## H.2 Ablation Study

### H.2.1 Initial and base learning rate

We conduct an ablation study to assess the impact of different choices for the base learning rate and the initial learning rate on training loss and test accuracy using ResNet-50.

**Initial learning rate** $\eta_0$ Our theory suggests that the choice of the initial $\eta_0$ will not influence the final loss performance, as long as $\eta_0$ is not too large. We tested this hypothesis by running each of the problems using values of $\eta_0$ ranging from $10^{-6}$ to 1. Figure 5 validates this conclusion in practice.

**Base learning rate** For this experiment alone, we consider Adam++ with different values of the base learning rate of $\eta_t = c \cdot \frac{\max_{i \leq t} \|\mathbf{x}_i - \mathbf{x}_0\|_2}{\sqrt{d}}$. According to our theory, our algorithms are expected to be unstable when $c > 1$ and slow to converge when $c < 1$. Figure 6 illustrates the performance around $c = 1$. In comparison, Figure 7 displays the results for AdamW optimizer. It can be observed that both Adam and Adam++ have a large stable range near the optimal base learning rate.

### H.2.2 Optimization and implicit regularization effects analysis

To investigate the optimization and implicit regularization effects of the proposed optimizers, we conducted experiments on a convex classification problem considered in (Zhang et al., 2024). The setup involves $n = 50$ training data points $\{(\mathbf{x}_i, y_i)\}_{i=1}^n$, where $\mathbf{x}_i \in \mathbb{R}^d$ ($d = 50$) and $y_i \in \{+1, -1\}$. The objective was to minimize the empirical loss $\mathcal{R}(\mathbf{w})$, which uses the logistic loss function $\ell(z) = \log(1 + e^{-z})$:$\mathcal{R}(\mathbf{w}) = \frac{1}{n} \sum_{i=1}^n \ell(\langle \mathbf{w}, y_i \cdot \mathbf{x}_i \rangle)$. We used a constant learning rate ($\eta$) of 1.0 for all optimizers, except for Adam, which used a tuned rate of $3 \times 10^{-3}$. The resulting normalized $\ell_\infty$- and $\ell_2$-margin curves are displayed in Figure 8. The results show that both versions of Adam++ achieve large $\ell_\infty$- and $\ell_2$-margins, surpassed only by Adam. This indicates that Adam++ offers a better balance between optimization and implicit regularization effects.

### H.2.3 Further investigation on Case 2 of Adam++

For Case 2 of Adam++, we propose a simplified version of the Adam step size calculation: $\mathbf{s}_t = \sqrt{(t+1) \cdot \mathbf{v}_t}$ instead of the original $\mathbf{s}_t = \sqrt{(t+1) \cdot \max_{k \leq t}(\mathbf{v}_k)}$. We tested this change on the CIFAR-10 image classification task using the ResNet-18 model and a constant learning rate schedule. The results are shown in Figure 9. The simplified version achieves a lower test loss but slightly lower test accuracy. This trade-off is consistent with numerous empirical observations (Gugger and Howard, 2018). Notably, this result suggests that the simplified version is a practical and viable alternative that enables faster training.

## H.3 Computational Overhead

In this section, we analyze computational overhead by plotting test loss and test accuracy against wall-clock time for different models with different scheduler on different datasets, as shown in

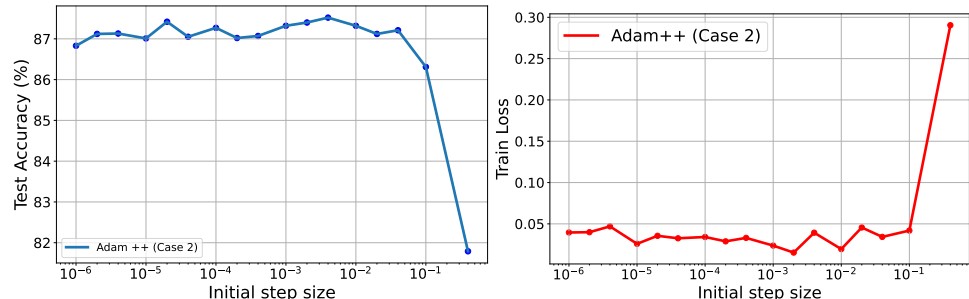

Figure 5: Effect of different choices of $\eta_0$ on test accuracy and training losses for Adam++ (Case 2). When $\eta_0$ is less than $10^{-1}$, its influence on final performance is marginal.

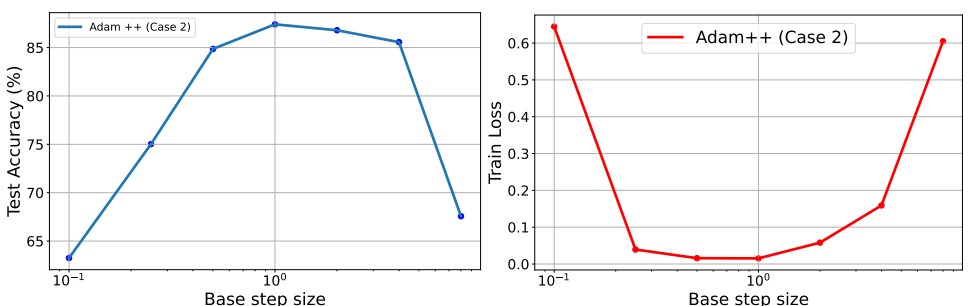

Figure 6: Effect of different choices of $c$ on test accuracy and training losses for Adam++ (Case 2). When $c$ is between $0.5$ and $4$, its influence on final performance is limited.

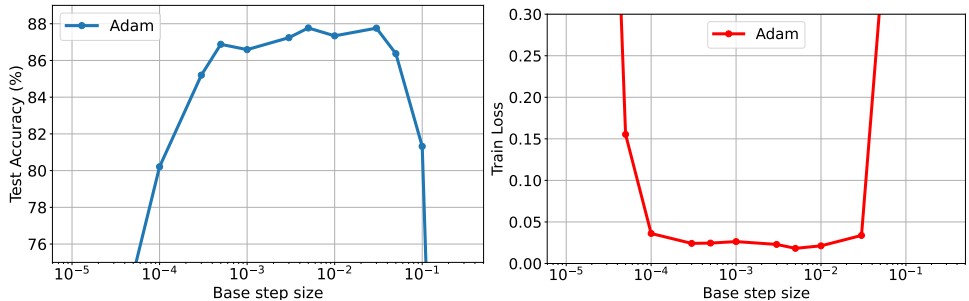

Figure 7: Effect of different choices of base step size (learning rate) on test accuracy and training losses for AdamW optimizer, where the base step size is searched between $1 \times 10^{-5}$ to $3 \times 10^{-1}$.

Figure 10. The results in Figure 10 are shown on the tasks of training ResNet-18 on CIFAR-10 with constant scheduler and training DenseNet-121 on CIFAR-10 with cosine scheduler. Adam++ Case 1 incurs less computational overhead compared to Prodigy and D-Adapt Adam, while delivering comparable or superior performance.

# I   DISCUSSION ON THE MEMORY USAGE OF ADAGRAD++ AND ADAM++

In this section, we briefly discuss the memory usage of AdaGrad++ and Adam++. Specifically, we note that, compared to vanilla AdaGrad and Adam, AdaGrad++ and Adam++ require the storage of an additional set of parameters, $\mathbf{x}_0$, resulting in slightly higher memory usage. However, it is important to highlight that, compared to existing parameter-free adaptive gradient methods such as Prodigy (Mishchenko and Defazio, 2024) and D-adaptation (Defazio and Mishchenko, 2023), which necessitate storing multiple intermediate quantities of the same size as the number of parameters, our proposed algorithms are more efficient in terms of memory usage.

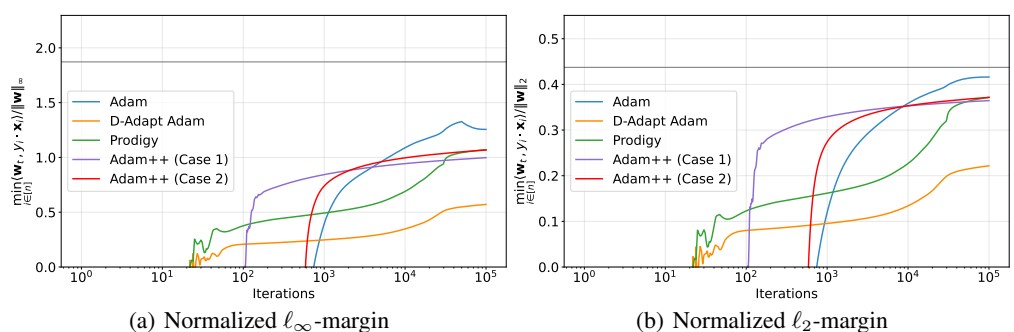

(a) Normalized $\ell_\infty$-margin

(b) Normalized $\ell_2$-margin

Figure 8: Normalized $\ell_\infty$-margins and $\ell_2$-margins achieved by for different optimizers, where the gray lines are the theoretically maximum margin values.

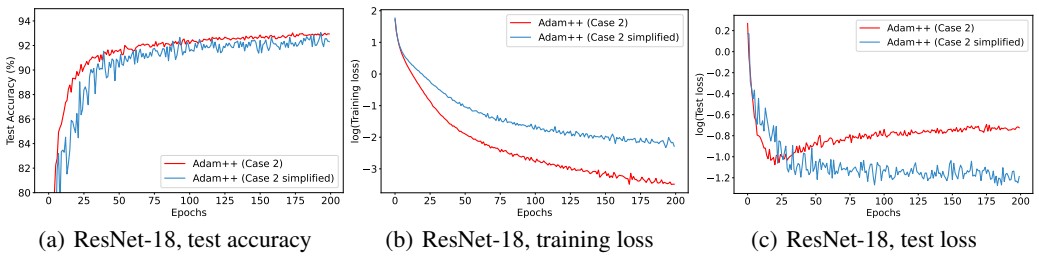

(a) ResNet-18, test accuracy

(b) ResNet-18, training loss

(c) ResNet-18, test loss

Figure 9: The results of training ResNet-18 on CIFAR-10 with Adam++ (Case 2) and its simplified version with a constant learning rate schedule.

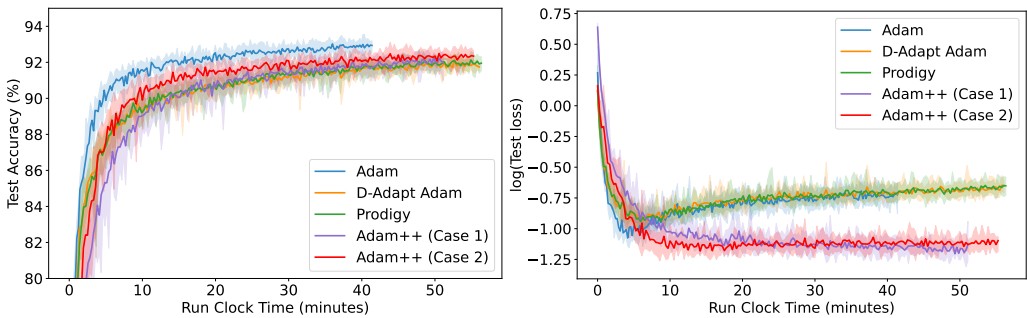

(a) ResNet-18, CIFAR-10, constant scheduler, test accuracy

(b) ResNet-18, CIFAR-10, constant scheduler, test loss

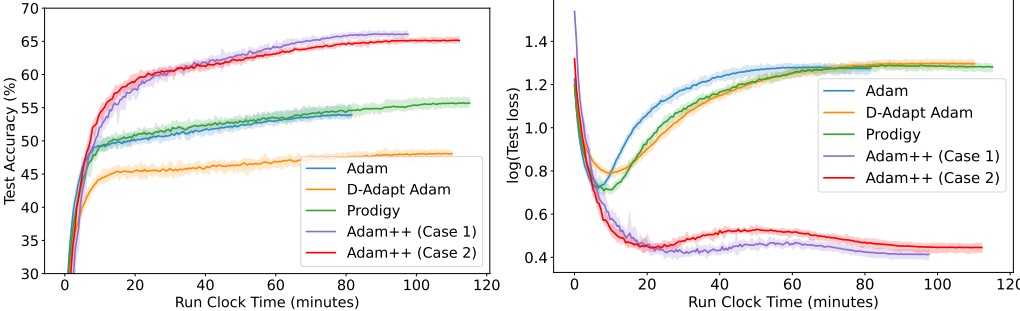

(c) DenseNet-121, CIFAR-100, cosine scheduler, test accuracy

(d) DenseNet-121, CIFAR-100, cosine scheduler, test loss

Figure 10: Test accuracy and test loss with respect to wall-clock time for different algorithms in training ResNet-18 and DenseNet-121 on CIFAR-10 and CIFAR-100 datasets with different schedulers.

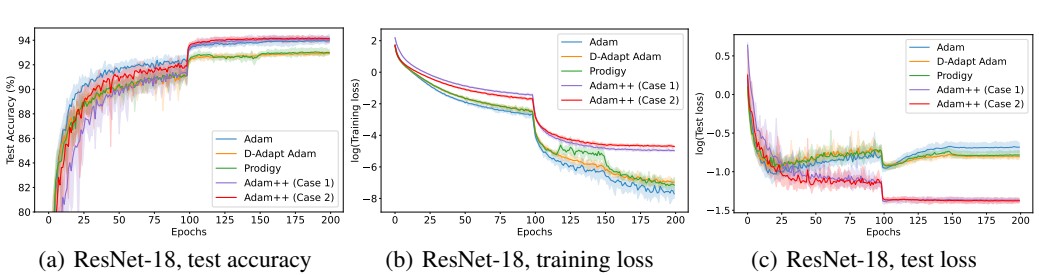

(a) ResNet-18, test accuracy    (b) ResNet-18, training loss    (c) ResNet-18, test loss

Figure 11: The results of training ResNet-18 on CIFAR-10 with a Multi-step learning rate schedule.

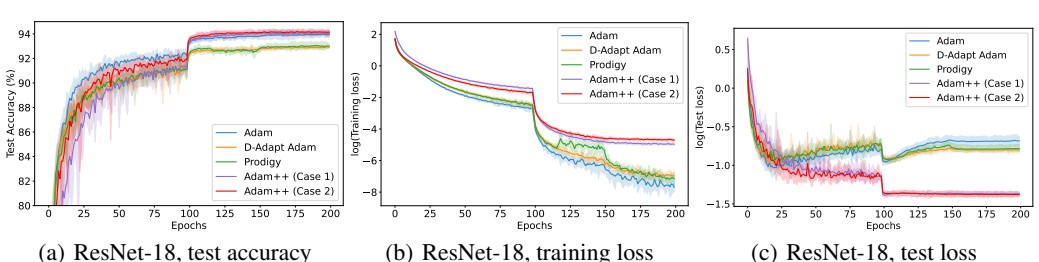

(a) ResNet-18, test accuracy    (b) ResNet-18, training loss    (c) ResNet-18, test loss

Figure 12: The results of training ResNet-18 on CIFAR-100 with a Multi-step learning rate schedule.

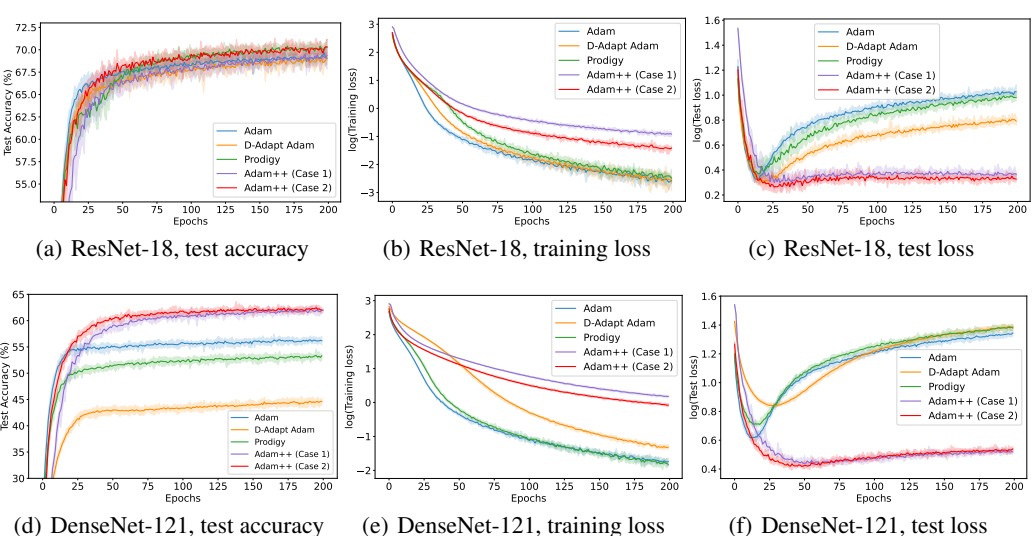

(a) ResNet-18, test accuracy    (b) ResNet-18, training loss    (c) ResNet-18, test loss

(d) DenseNet-121, test accuracy    (e) DenseNet-121, training loss    (f) DenseNet-121, test loss

Figure 13: The results of training ResNet-18 and DenseNet-121 on CIFAR-100 with a constant learning rate schedule.

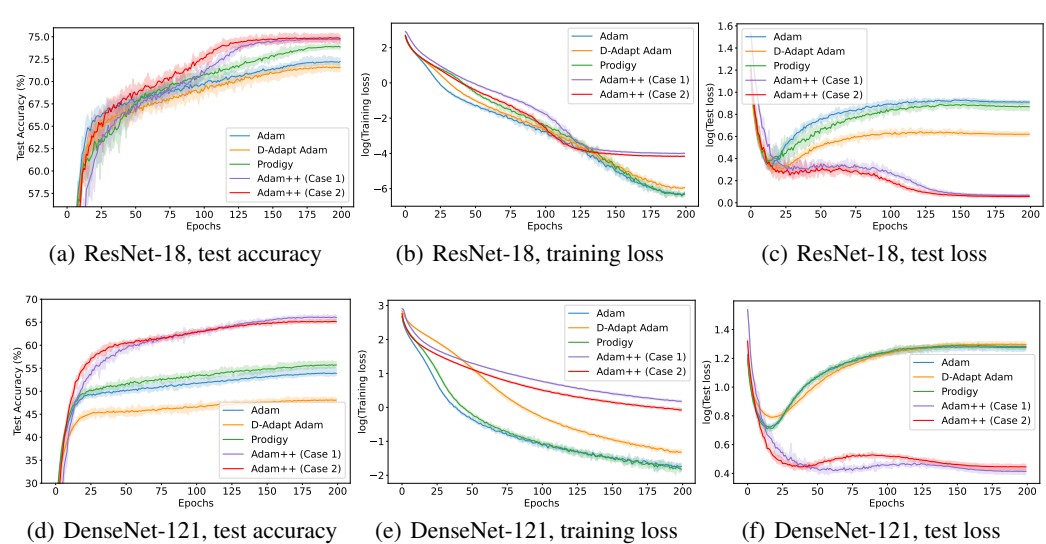

(a) ResNet-18, test accuracy    (b) ResNet-18, training loss    (c) ResNet-18, test loss

(d) DenseNet-121, test accuracy    (e) DenseNet-121, training loss    (f) DenseNet-121, test loss

Figure 14: The results of training ResNet-18 and DenseNet-121 on CIFAR-100 with a cosine learning rate schedule.

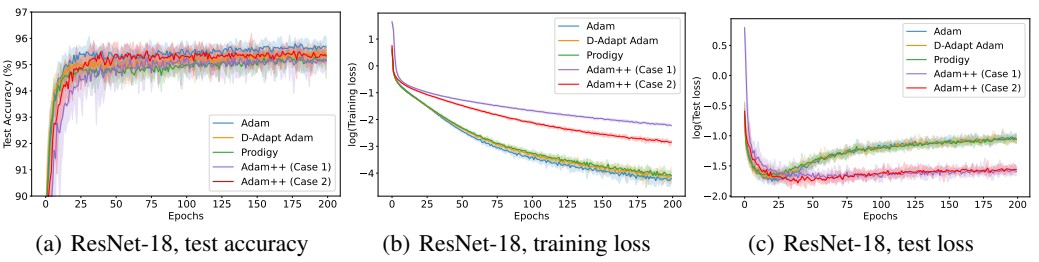

(a) ResNet-18, test accuracy    (b) ResNet-18, training loss    (c) ResNet-18, test loss

Figure 15: The results of training ResNet-18 on SVHN with a constant learning rate schedule.

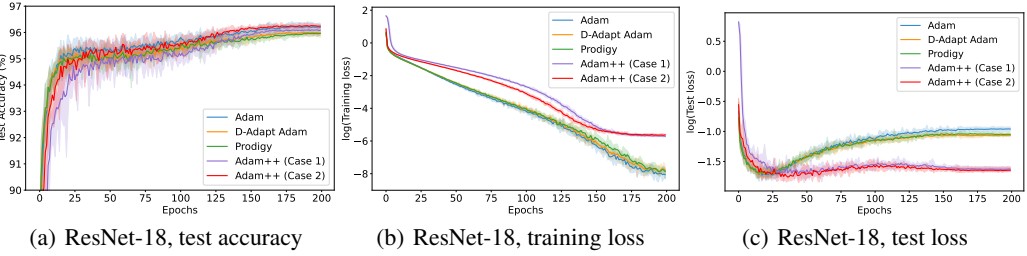

(a) ResNet-18, test accuracy    (b) ResNet-18, training loss    (c) ResNet-18, test loss

Figure 16: The results of training ResNet-18 on SVHN with a cosine learning rate schedule.

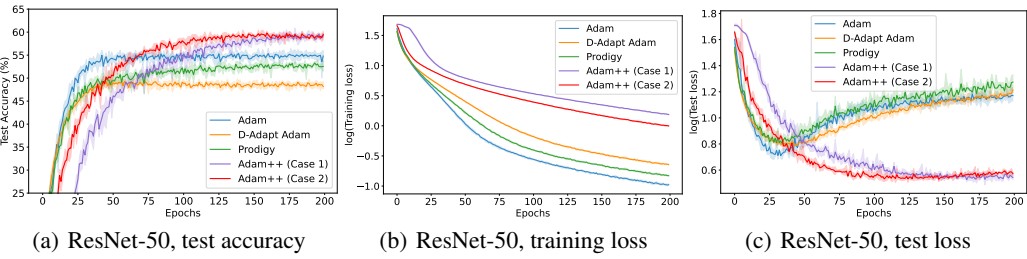

(a) ResNet-50, test accuracy    (b) ResNet-50, training loss    (c) ResNet-50, test loss

Figure 17: The results of training ResNet-50 with on Tiny-ImageNet with a constant learning rate schedule.

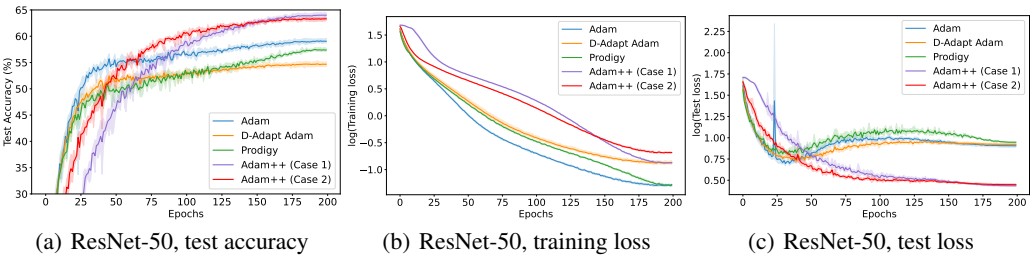

(a) ResNet-50, test accuracy    (b) ResNet-50, training loss    (c) ResNet-50, test loss

Figure 18: The results of training ResNet-50 on Tiny-ImageNet with a cosine learning rate schedule.

