# OpenReview forum: "Towards Simple and Provable Parameter-Free Adaptive Gradient Methods"
_ICLR.cc/2026/Conference — Submitted to ICLR 2026_

### Official Review · Reviewer_6s59 · 2025-10-29

**Soundness:** 2
**Presentation:** 2
**Contribution:** 2
**Rating:** 4
**Confidence:** 3

**Summary:**

This paper proposes a novel parameter-free adaptive gradient method, scaling the distance over gradient by $1/\sqrt{d}$ for AdaGrad++ and by $1/\sqrt{d(t+1)}$ for Adam++. Theoretical convergence results, along with empirical studies, are presented.

**Strengths:**

1. The idea is simple.

2. Theoretical convergence results, with generalized assumptions, are presented.

**Weaknesses:**

1. The paper is easy to follow since the idea is simple. However, the writing is rough and lacks careful checking and polishing.

2. It should be explained why $\eta_t$ should involve a max operation to ensure its nondecreasing.

3. Though it is a common issue, this method still requires a base learning rate, which needs to be tuned.

4. The results in Figures 1 and 2 indicate that the improvement in test accuracy achieved by Adam++ primarily stems from its reduced overfitting -- the baseline models exhibit overfitting; the level of overfitting and underfitting can be adjusted by modifying the (base) learning rate; this makes the results less convincing.

**Questions:**

1. Why is a nondecreasing $\eta_t$ preferred?

---

> ### Author Response · Authors · 2025-11-21
>
> Thank you for your concurrence with the main argument of our paper, and we have carefully considered your suggestions and provided detailed responses below:
>
> > Q1: The paper is easy to follow since the idea is simple. However, the writing is rough and lacks careful checking and polishing.
>
> A1: Thank you for your careful inspection of the writing of our paper. We are sorry for this and we have gone through proof-reading and fixed typos in the revised version of our paper.
>
> > Q2: It should be explained why $\eta_t$ should involve a max operation to ensure its nondecreasing. Why is a nondecreasing $\eta_t$ preferred?
>
> A2: A nondecreasing $\eta_t$ is required in the proof of the proposed algorithms (such as Lemmas E.1 and E.2). Actually, $\eta_t$ remains almost unchanged after the initial steps of training. Therefore, it can be seen as a constant learning rate scheduler near the optimal point.
>
> > Q3: Though it is a common issue, this method still requires a base learning rate, which needs to be tuned.
>
> A3: Thank you for the question. It is important to note that previous parameter-free algorithms, such as DoG [1], still require tuning certain hyperparameters, including the initial step size. As detailed in Appendices F.3 and F.4 of [1], the default settings can even lead to training failure. In contrast, we present extensive ablation studies in Appendix H.2 regarding the sensitivity of our method. These experiments demonstrate that our algorithms maintain near-optimal performance across a wide range of hyperparameter settings, confirming their robustness.
>
> > Q4: The results in Figures 1 and 2 indicate that the improvement in test accuracy achieved by Adam++ primarily stems from its reduced overfitting -- the baseline models exhibit overfitting; the level of overfitting and underfitting can be adjusted by modifying the (base) learning rate; this makes the results less convincing.
>
> A4: We optimized the hyperparameters for all algorithms by performing a parameter sweep. Specifically, the AdamW learning rate was searched within $ [1\times 10^{-4}, 1\times 10^{-2}] $ and other optimizers between $ [0.1, 3.0] $, selecting the setting that achieved the best accuracy. Appendix H.2 includes ablation studies which confirm the robustness of our algorithms across a wide hyperparameter range.
>
> [1] Ivgi, Maor, Oliver Hinder, and Yair Carmon. "DoG is SGD’s best friend: A parameter-free dynamic step size schedule." *International Conference on Machine Learning*. PMLR, 2023.

---

> ### Comment · Reviewer_6s59 · 2025-11-27
>
> Thanks for the response.
>
> If $\eta_t$ remains almost unchanged after the initial steps of training, would it imply that the defined term does not have adaptivity after the initial steps? It then appears strange. If it is reduced to a constant after a short period, why would it perform better than a constant learning alternative? Since it has a warm-up effect? Then, what if we introduce a, e.g., linear, warm-up for the others?
>
> Besides, an intuitive explanation of why the distance between the current parameter and the initial parameters can reasonably serve as a proxy for the learning rate could be interesting and helpful. In particular, why should the learning rate be larger when the distance gets larger? This appears counterintuitive.
>
> Even if all methods are well-tuned, the concern that the improved test accuracy appears to stem from its reduced overfitting is not fully addressed. Why would it tend to have reduced overfitting? Does it imply the learning rate is being relatively too large or too small? This might be the intrinsic cause. You may visualize curves of the effective learning rate for a clear comparison.

---

> > ### Author Response · Authors · 2025-12-03
> >
> > Thank you for your further questions. We answer them as follows.
> >
> > >Re: If $\eta_t$ remains almost unchanged after the initial steps of training, would it imply that the defined term does not have adaptivity after the initial steps? It then appears strange. If it is reduced to a constant after a short period, why would it perform better than a constant learning alternative? Since it has a warm-up effect? Then, what if we introduce a, e.g., linear, warm-up for the others?
> >
> > We would like to clarify that the effective learning rate of Adam++ is $\frac{\eta_t}{\sqrt{H_t}}$, where $H_t$ is the second moment of the gradients and contributes to the adaptivity of the learning rate. So even if $\eta_t$ becomes almost unchanged after a certain number of steps, $H_t$ still changes and adaptivity remains.
> >
> > >Re: an intuitive explanation of why the distance between the current parameter and the initial parameters can reasonably serve as a proxy for the learning rate could be interesting and helpful. In particular, why should the learning rate be larger when the distance gets larger? This appears counterintuitive.
> >
> > We appreciate the reviewer’s request for a more intuitive explanation. Our step–size rule is not based on the heuristic claim that “the learning rate should increase whenever the distance from the initialization increases”. Instead, the distance term appears naturally when one tries to optimize the standard SGD regret bound without knowing the problem scale in advance.
> > In the convex analysis, the excess risk bound consists of two terms:
> >
> > $$\text{optimization term} \propto \frac{||x^*−x_0||}{\eta}$$
> > and
> > $$\text{noise term} \propto \eta \sum_{t=1}^T=||g_t||^2$$
> >
> > so that the bound is minimized by a step–size of the form
> > $$η^* \propto \frac{||x^∗−x_0||}{\sum_{t=1}^T ||g_t||^2}$$
> >
> > Thus, even in the classical theory, a larger distance between the initialization and the optimum $||x^∗−x_0||$ calls for a larger optimal step–size.
> >
> > The difficulty is that $||x^∗−x_0||$ is unknown. Our analysis shows that, on trajectories where the iterates remain in a ball $B(x_0,R)$ containing $x^∗$, the maximal observed distance
> > $$d_t  =  \max_{⁡0\le i\le t}||x_i−x_0||$$
> > serves as an on-the-fly surrogate on this unknown radius. Replacing $||x^∗−x_0||$ by $d_t$ yields exactly the DoG/Adam++ rule
> > $η_t  =  \frac{d_t}{\sqrt{\sum_{i=1}^t||g_i||^2}}$.
> > This is why the DoG/Adam++ learning rate is increasing in $d_t$: it is simply tracking the problem’s effective distance scale implied by the current trajectory, while the denominator $\sqrt{\sum_{i=1}^t||g_i||^2}$ ensures that large or noisy gradients still shrink the step size.

---

### Official Review · Reviewer_hSug · 2025-10-30

**Soundness:** 3
**Presentation:** 3
**Contribution:** 3
**Rating:** 4
**Confidence:** 4

**Summary:**

This paper proposes new optimizers named AdaGrad++ and Adam++, which are parameter-free variants of AdaGrad and Adam, respectively.
They can be implemented with a small modification to AdaGrad and Adam, whereas existing parameter-free methods require complicated implementations.
The authors also provide theoretical proofs that ensure the convergence of their optimizers in the convex optimization setting.
In the experiments, they show that Adam++ can achieve competitive performance with Adam without the need of learning rate tuning in image classification tasks.

**Strengths:**

- Their proposed optimizers can be implemented very easily by slightly modifying the original AdaGrad and Adam.
- Presentation of the paper is clear and easy to read.
- Although existing parameter-free optimizers tend to underperform well-tuned Adam, Adam++ seems to work well, showing competitive performance with the original Adam without learning-rate tuning.
- As far as I checked, there were no mathematical flaws in the proofs of their theorems.

**Weaknesses:**

### Experimental settings are limited to image classification

I would like to see results in other settings, such as language modeling, reinforcement learning, generative modeling (e.g., diffusion models), etc.

### Lack of Theoretical Results in Non-convex settings

Theoretical analysis is only provided for convex settings, but there are a lot of convergence analysis of Adam/AdaGrad in smooth non-convex settings.
These optimizers are mainly used for deep learning models, whose objective functions are highly non-convex, so it is desirable that the analysis in non-convex settings is included (see [1, 2, 3] for example).

### Gap between theory and practice

In the derivation of Adam++, the authors use AMSGrad's update rule to ensure the theoretical convergence, but they omit it and use a simplified version in the practical implementation, which degrades the soundness of theoretical results.

### Additional memory costs

In the proposed method, the initial parameter $mathbf{x}_0$ has to be stored to calculate $r_t$, which requires additional memory in addition to the other optimizer states (e.g., $\mathbf{m}_t$ and $\mathbf{v}_t$ in Adam).
This can be a disadvantage in the era of large-scale models.

### Minor comments

- The authors should cite [4] in Section 3, which also proposes a method to guarantee the convergence of Adam with a small modification.
- The authors do not include "LLM Usage" statements in the paper, which is mandatory as described in the official guideline.

[1] Chen, Xiangyi, et al. "On the convergence of a class of adam-type algorithms for non-convex optimization." arXiv preprint arXiv:1808.02941 (2018).

[2] Défossez, Alexandre, et al. "A simple convergence proof of adam and adagrad." arXiv preprint arXiv:2003.02395 (2020).

[3] Zhang, Yushun, et al. "Adam can converge without any modification on update rules." Advances in neural information processing systems 35 (2022): 28386-28399.

[4] Taniguchi, Shohei, et al. "ADOPT: Modified Adam Can Converge with Any $\beta_2 $ with the Optimal Rate." Advances in Neural Information Processing Systems 37 (2024): 72438-72474.

**Questions:**

- In my understanding, Case 1 of Adam++ is identical to AdaGrad++. Is this correct? If not, what is the difference between them?
- The authors use AMSGrad modification to ensure the convergence of Adam++, but AMSGrad tends to slow down the convergence in practice. How about using the technique of ADOPT [4], which can also ensure the theoretical convergence by slightly modifying the original Adam. To my knowledge, ADOPT works better than AMSGrad and Adam in many cases, so it may help bridge the gap between theory and practice.

[4] Taniguchi, Shohei, et al. "ADOPT: Modified Adam Can Converge with Any $\beta_2 $ with the Optimal Rate." Advances in Neural Information Processing Systems 37 (2024): 72438-72474.

---

> ### Author Response · Authors · 2025-11-21
>
> Thank you for your detailed and thoughtful review. Below, we provide detailed responses to each of your comments.
>
> > Q1: Experimental settings are limited to image classification: I would like to see results in other settings, such as language modeling, reinforcement learning, generative modeling (e.g., diffusion models), etc.
>
> A1: Thank you for your suggestion. We provided the results of language modeling tasks with LLMs including GPT-2 small and medium in Section 6.2 The results in Figures 3 and 4 show that our algorithms can handle more general scenarios. And we will implement more empirical study in other settings in the future.
>
> > Q2: Lack of Theoretical Results in Non-convex settings: Theoretical analysis is only provided for convex settings, but there are a lot of convergence analysis of Adam/AdaGrad in smooth non-convex settings. These optimizers are mainly used for deep learning models, whose objective functions are highly non-convex, so it is desirable that the analysis in non-convex settings is included (see [1, 2, 3] for example).
>
> A2: Thank you for your suggestion. We provided the analysis and proof for convergence in the nonconvex optimization setting of our algorithms in Appendix D. We have added more details to help comprehension in the revised paper.
>
> > Q3: Gap between theory and practice: In the derivation of Adam++, the authors use AMSGrad's update rule to ensure the theoretical convergence, but they omit it and use a simplified version in the practical implementation, which degrades the soundness of theoretical results.
>
> A3: We are sorry that we did not clarify the application of the "simplified" version of Case 2 of Adam++. Actually, we only apply $\mathbf{s}_t=\sqrt{(t+1)\mathbf{v}_t}$ for language modeling tasks and use AMSGrad-like update rule (the original algorithm) for other experiments. And we added this clarification in Section 5.1.
>
> > Q4: Additional memory costs: In the proposed method, the initial parameter $\mathbf{x}_0$ has to be stored to calculate $\mathbf{r}_t$, which requires additional memory in addition to the other optimizer states (e.g., $\mathbf{m}_t$ and $\mathbf{v}_t$ in Adam). This can be a disadvantage in the era of large-scale models.
>
> A4: We acknowledge that our algorithm may require more memory than base algorithms of Adagrad and Adam. However, Deepspeed's ZeRO Offload technique [1] solves this issue by distributing the optimizer states to different devices. Therefore, in the setting of large-scale distributed training, when the number of GPUs scales up, the additional usage of memory is quite minimal.
>
> > Q5: The authors should cite [4] in Section 3, which also proposes a method to guarantee the convergence of Adam with a small modification.
>
> A5: We are sorry for missing the citation of ADOPT paper. We have added it and discussed it in the related work section in the revision of our paper.
>
> > Q6: The authors do not include "LLM Usage" statements in the paper, which is mandatory as described in the official guideline.
>
> A6: Thank you for your suggestion. We only use LLM in our language modeling tasks since we need to compare optimization methods in training GPT-2 small and medium models in our experiments. We have added LLM Usage statements before Reference section in the revised paper.
>
> > Q7: In my understanding, Case 1 of Adam++ is identical to AdaGrad++. Is this correct? If not, what is the difference between them?
>
> A7: We appreciate your interest in the difference in our algorithms. Actually, Case 1 of Adam++ introduces momentum to the gradient to stabilize the training, which is one of the core features of a series of algorithms, including Adam and AdamW, which contributes to better performances than approaches without momentum like SGD.

---

> ### Author Response · Authors · 2025-11-21
>
> > Q8: The authors use AMSGrad modification to ensure the convergence of Adam++, but AMSGrad tends to slow down the convergence in practice. How about using the technique of ADOPT [4], which can also ensure the theoretical convergence by slightly modifying the original Adam. To my knowledge, ADOPT works better than AMSGrad and Adam in many cases, so it may help bridge the gap between theory and practice.
>
> A8: Thank you for your introduction of the new idea that combines ADOPT into our framework. We attempted to introduce the techniques of ADOPT as follows:
>
> ---
>
> Input $\mathbf{x}_0,\eta_0,\beta_1,\beta_2,\lambda$
>
> $\mathbf{v}_0\leftarrow \mathbf{g}_0\odot\mathbf{g}_0,~\mathbf{m}_1\leftarrow\mathbf{g}_1/\max(\sqrt{\mathbf{v}_0},\epsilon)$
>
> For t=1 to T do
>
> $\qquad r_t=\|\mathbf{x}_t-\mathbf{x}_0\|/\sqrt{d}$
>
> $\qquad\eta_t=\max(\eta_{t-1},r_t)$
>
> $\qquad\mathbf{g}_t=\mathcal{G}(\mathbf{x}_t)$
>
> $\qquad\mathbf{x}_t=\mathbf{x}\_{t-1}-\eta_t(\mathbf{m}\_t+\lambda\mathbf{x}\_{t-1})$
>
> $\qquad\mathbf{v}\_t=\beta_2\mathbf{v}\_{t-1}+(1-\beta_2)\mathbf{g}_t\odot\mathbf{g}_t$
>
> $\qquad\mathbf{m}\_{t+1}=\beta_1\mathbf{m}\_t+(1-\beta\_1)\text{Clip}(\frac{\mathbf{g}\_{t+1}}{\max(\sqrt{\mathbf{v}_t},\epsilon)},c_t)$
>
> End for
>
> Return $\mathbf{x}_t$
>
> ---
>
> We set $c_t\equiv1$, $\lambda=5e-4$, $(\beta_1,\beta_2)=(0.9,0.999)$, $\eta_0=1e-6$, and do experiments on the CIFAR10 dataset with the ResNet18 model with a constant learning rate scheduler. The accuracy once achieved 70%. but later the gradient exploded. We will investigate this variant of ADOPT further in future work. In addition, we are not sure if the theoretical convergence rates also hold for the parameter-free version of ADOPT, which is an interesting future work too. We have added ADOPT into the related work of our revised paper.
>
> [1] Rajbhandari, Samyam, et al. "Zero: Memory optimizations toward training trillion parameter models." *SC20: International Conference for High Performance Computing, Networking, Storage and Analysis*. IEEE, 2020.

---

### Official Review · Reviewer_66vf · 2025-10-30

**Soundness:** 3
**Presentation:** 3
**Contribution:** 3
**Rating:** 6
**Confidence:** 4

**Summary:**

The paper proposes two "parameter-free" optimization algorithms named AdaGrad++ and Adam++.  Parameter-free here means that one does not need to sweat the implicit or explicit parameters of the algorithm because the convergence is robust to their values. AdaGrad++ can be seen as a slight extension of the algorithm of Ivgi. Adam++ adapts Ivgi's idea to the Adam algorithm.

The authors provide convergence proofs for both algorithms. Although providing such proof is better than what certain competing algorithms can offer, these proofs give very little insight about the parameter-free nature of the algorithms, that is, about of the robustness of their performance across the parameter range (unlike Ward's paper, for instance).

Instead one has to rely on empirical evidence to convince oneself that parameter searching is a thing of the past.  I would have liked experiments showing how robust the result when one multiplies the suggested learning rates by some constants. Nevertheless, the experiments are insightful because they show the evolution of both the training and testing costs for diverse algorithms, making clear that this is not just a matter of optimization, but also of implicit regularization.

**Strengths:**

- clear statement and valuable proofs
- generally good empirical results

**Weaknesses:**

- no empirical assessment of the robustness of the proposed algorithms against parameter changes (or rescalings).
- proofs say very little about the parameter free nature of the proposed algorithms (too hard maybe?).
- no clear discussion of the optimization effects vs the implicit regularization effects.

- (minor) lots of distracting typos "entry-wisely" "AdaGard"

**Questions:**

- did you run experiments to assess the robustness of the proposed algorithms against parameter changes (or rescalings)?

---

> ### Author Response · Authors · 2025-11-21
>
> Thank you for your careful evaluation of our work. We appreciate your recognition of our contributions and your constructive questions. We have carefully considered your concerns and we provide detailed responses to your questions below.
>
> > Q1: no empirical assessment of the robustness of the proposed algorithms against parameter changes (or rescalings).
> >
> > did you run experiments to assess the robustness of the proposed algorithms against parameter changes (or rescalings)?
>
> A1: Thank you for your question. We investigated the effect of initial and base learning rates of Adam++ in Appendix H.2.1. To be more comparable, we added the curves for Adam optimizer with different learning rates in Figure 7 in the revised version of our paper. As the sensitivity curves show, Adam++ has a large stable range near the optimal initial step size and base step size.
>
> > Q2: proofs say very little about the parameter free nature of the proposed algorithms (too hard maybe?).
>
> A2: We are sorry that we did not make it clear to the parameter-free nature of our algorithms. Actually, according to (B.3), the bound on f(x) can be divided into 3 parts: $I_1$, the improvement of Bregman divergence between the current parameter to the optimal parameter; $I_2$, the weighted sum of the gradients adjusted by $\eta_t$; and the noise term, which is also adjusted by $\eta_t$. The adjustment of automatically computed step size $\eta_t$is the core of parameter-free nature: with $\eta_t$, $I_2$ can be bounded by $O(\sqrt{T})$ and the noise term can be bounded by $O(\sqrt{T})$ with high probability as well. We added more explanation to the proof for better comprehension of the parameter-free nature.
>
> > Q3: no clear discussion of the optimization effects vs the implicit regularization effects.
>
> A3: Thank you for your suggestion. To further investigate these effects, we added the experiments in convex setting proposed by [1] in Appendix H.2.2. The experiment results show that our algorithm can actually achieve near optimal value in convex setting. Taking other experiments in nonconvex settings into account, the discussions above demonstrate that our algorithm can balance the optimization effects and the implicit regularization effects well. However, Assumption 4.3 in [1] requires that the limitation of step size is 0, which does not hold for our algorithms. But we suspect this issue since it may not be necessary.
>
> > Q4: (minor) lots of distracting typos "entry-wisely" "AdaGard"
>
> A4: Thank you for your careful investigation into our paper. We are sorry to have some typos in our paper. We went through a proof-read and fixed these typos in the revised paper.
>
> [1] Zhang, Chenyang, Difan Zou, and Yuan Cao. "The implicit bias of adam on separable data." *Advances in Neural Information Processing Systems* 37 (2024): 23988-24021.

---

### Official Review · Reviewer_J4sP · 2025-11-07

**Soundness:** 2
**Presentation:** 3
**Contribution:** 1
**Rating:** 2
**Confidence:** 3

**Summary:**

The submission presents a combination of parameter-free methods based on DOG (small dependence on initial step-size) and adaptive methods like AdaGrad and Adam (the AMSGrad variant) to obtain new variants, AdaGrad++ and Adam++.
The submission provides convergence rates for the proposed methods and shows that they perform similarly to Adam on standard benchmarks.

**Strengths:**

The problem of finding optimizers that work out of the box and do not require tuning is important and relevant to the ICLR community. The resulting algorithm is indeed simple, and the presentation of the algorithms is clear.

**Weaknesses:**

My main issues with the submissions
are that the theoretical guarantees appear weak,
and that the motivation for the proposed method is unclear.

**Theoretical guarantees are weak,**
or I have a very hard time understanding them.
The submission claims that the rates in Thm 4.2 and 5.1 are at the worst-case sublinear rate because $\vert s \vert$ grows in $\sqrt{T}$ in the worst case,
but this seems to ignore the dependence on other quantities that depend on $t$,
such as the maximum distance to the optimum $D$
and the step-size $\eta_T$, as the step-size is set as a function of $x_t - x_0$.
The submission does not discuss what happens to those quantities,
and as a result I do not see how the rates can be interpreted as the claimed $1/\sqrt{T}$ rates.

**Improvement over prior work is unclear.**
There is a large body of work on parameter-free, adaptive optimizers and their combination, and it is not clear to me what the main advantages of the proposed method are.
The related work section cites many papers,
but does not build a compelling case for a hole in the literature that the submission fills.
The combination of D-Adaptation and AdaGrad already provided a parameter-free with coordinate-wise adaptive method, and the submission does not make a strong case for why the proposed AdaGrad++ is better.
Although this might be partially due to the
fact that I do not understand the theoretical guarantees.


Minor issues (no response required):

- The font in the figures is very small and not readable.
- The figure showing the sensitivity to initial step-size for Adam++ in Figure 5 is not very informative without a comparison with plain Adam
- Please fix the references.
  Many are duplicated, refer to arxiv versions instead of published versions, don't have years,

**Questions:**

Why are the theoretical guarantees valid convergence rates that give a $O(1/\sqrt{T})$ rate?

How do those rates compare to the actual rates on problems of interest, say convex Lipschitz? What do we loose from not having to specifiy $G$ and $D$ in advance?

---

> ### Author Response · Authors · 2025-11-21
>
> Thank you for your detailed and thoughtful review, and we sincerely appreciate your recognition of our work. Moreover, your detailed questions have been invaluable in helping us refine our work, and we listed our responses to your questions below.
>
> > Q1: **Theoretical guarantees are weak,** or I have a very hard time understanding them. The submission claims that the rates in Thm 4.2 and 5.1 are at the worst-case sublinear rate because $\vert s \vert$ grows in $\sqrt{T}$ in the worst case, but this seems to ignore the dependence on other quantities that depend on $t$, such as the maximum distance to the optimum $D$ and the step-size $\eta_T$, as the step-size is set as a function of $x_t-x_0$.  The submission does not discuss what happens to those quantities, and as a result I do not see how the rates can be interpreted as the claimed $1/\sqrt{T}$ rates.
> >
> > Why are the theoretical guarantees valid convergence rates that give a $O(1/\sqrt{T})$ rate?
>
> A1: Thank you for point out this issue. We are sorry that we didn't make this point clear in the original version. As discussed in the DoG[1], in practice, $\bar{D}\_\tau=\max\_{t\le\tau}\|\mathbf{x}\_t-\mathbf{x}^\*\|\_2$ is often bounded by $O(\|\mathbf{x}\_0-\mathbf{x}^\*\|\_2)$, e.g., $\bar{D}\_\tau \leq 3 \|\mathbf{x}\_0-\mathbf{x}^\*\|\_2$, so the iterates do not move too far away from $\mathbf{x}\_0$. With this additional condition,$\eta\_T$ is bounded by $O(\|\mathbf{x}\_0-\mathbf{x}^\*\|\_2/d)$, and $D\_\tau=\max\_{t\le\tau}\|\mathbf{x}\_t-\mathbf{x}^*\|\_\infty$ is also bounded by $O(\|\mathbf{x}\_0-\mathbf{x}^\*\|\_2)$. In addition, DoG[1] also proposed a step size scheme whose iterates are guaranteed to remain bounded with high probability (See Section 3.3 and Proposition 2 in their paper). By using a similar step size scheme, we can also develop modified versions of AdaGrad++ and Adam++, such that the distance between $\|\mathbf{x}\_t-\mathbf{x}^\*\|\_2$ will be bounded by $O(\|\mathbf{x}\_0-\mathbf{x}^\*\|\_2)$ with high probability. Therefore, in our discussion, we will assume $\bar{D}\_\tau$ is bounded by $O(\|\mathbf{x}\_0-\mathbf{x}^\*\|\_2)$ and $M\_1,M\_2$ and $M\_3$ are some constants up to some logarithmic factors. Since $\|\mathbf{s}\_\tau\|\_2$ is bounded by $\sqrt{T}$, we achieved $\tilde{O}(1/\sqrt{T})$ convergence rate for the proposed algorithms. We have added a detailed discussion and explanation in the revised paper.
>
>
> > Q2: **Improvement over prior work is unclear. **There is a large body of work on parameter-free, adaptive optimizers and their combination, and it is not clear to me what the main advantages of the proposed method are. The related work section cites many papers, but does not build a compelling case for a hole in the literature that the submission fills. The combination of D-Adaptation and AdaGrad already provided a parameter-free with coordinate-wise adaptive method, and the submission does not make a strong case for why the proposed AdaGrad++ is better. Although this might be partially due to the fact that I do not understand the theoretical guarantees.
>
> A2: Thank you for your question. Our primary motivation is to propose parameter-free algorithms that are significantly simpler than existing baselines while maintaining a comparable convergence rate of $O(1/\sqrt{T})$. We demonstrate that achieving this parameter-free property requires only minimal modifications to the standard Adagrad and Adam algorithms. In contrast, methods like D-Adapt AdamW [2] currently lack theoretical convergence guarantees. Nevertheless, we acknowledge the foundational algorithmic and theoretical innovations introduced by prior works, including DoG [1] and D-Adaptation [2], upon which our work builds.
>
> > Q3: The font in the figures is very small and not readable.
>
> A3: We have enlarged the sizes of words in the figures in the revised paper.
>
> > Q4: The figure showing the sensitivity to initial step-size for Adam++ in Figure 5 is not very informative without a comparison with plain Adam
>
> A4: Thank you for your suggestion. We have added Figure 7 in Appendix H.2 as a reference to the sensitivity of Adam optimizer. As the sensitivity curves show, Adam++ has a large stable range near the optimal initial step size and base step size.
>
> > Q5: Please fix the references. Many are duplicated, refer to arxiv versions instead of published versions, don't have years,
>
> A5: Thank you for your meticulous inspection of our paper. We have fixed these references in the revised version of our paper.

---

> ### Author Response · Authors · 2025-11-21
>
> > Q6: How do those rates compare to the actual rates on problems of interest, say convex Lipschitz? What do we loose from not having to specifiy $G$ and $D$ in advance?
>
> A6: The constants such as $\|\mathbf{x}\_0-\mathbf{x}^*\|\_2$ and $L$ (the bound of gradient $\mathcal{G}(\mathbf{x}\_t)$) are only required for proving the convergence, and are not required for applying these in implementing the proposed algorithms. In addition, as we have explained in A1, the convergence rates of AdaGrad++ and Adam++ match that of AdaGrad and Adam for convex and Lipscthiz optimization problems.
>
> [1] Ivgi, Maor, Oliver Hinder, and Yair Carmon. "DoG is SGD’s best friend: A parameter-free dynamic step size schedule." *International Conference on Machine Learning*. PMLR, 2023.
>
> [2] Defazio, Aaron, and Konstantin Mishchenko. "Learning-rate-free learning by d-adaptation." *International Conference on Machine Learning*. PMLR, 2023.

---

### Author Response · Authors · 2025-11-21
**General Response to Reviewers and Area Chairs**

We sincerely thank the reviewers for their careful reading of our work and their detailed, constructive comments. In this revision, we have thoroughly proofread the manuscript to correct typos, fix the references (Q5 of Reviewer J4sP), and have enlarged the figure legends for better readability (Q4 of Reviewer J4sP). Furthermore, we have expanded the explanations of our proposed theories, conducted experiments on hyperparameter robustness, investigated Case 2 of Adam++ in greater depth, and analyzed the optimization and implicit regularization effects. All new materials are marked in blue within the revised paper.

### **1. Theoretical Clarifications**

We have enhanced the discussion regarding convergence rates in Section 4.1 and Section 5.2 (Q1 of Reviewer J4sP). Additionally, we clarified the distinctions between the AMSGrad-style and non-AMSGrad-style variants of Adam++ Case 2 in Section 5.1 and Appendix H.2.3 (Q3 of Reviewer hSug). Finally, we provided additional explanatory details for the proofs located in Appendices B, C, and D (Q2 of Reviewer 66vf).

### **2. Extended Experimental Evaluation**

We implemented additional ablation studies to empirically validate the optimization and implicit regularization effects in Appendix H.2.2 (Q3 of Reviewer 66vf), as well as experiments comparing Case 2 of Adam++ with and without the AMSGrad-style mechanism in Appendix H.2.3 (Q3 of Reviewer hSug).

To further illustrate the robustness of our proposed method, we added baseline figures for AdamW across different step sizes in Figure 7 (Q4 of Reviewer J4sP and Q1 of Reviewer 66vf).

Additionally, we relocated the language model experiments from Appendix H to Section 6.2 to present a comprehensive empirical validation of the proposed optimizers (Q1 of Reviewer hSug).

### **3. Reference Updates**

We have added ADOPT (Taniguchi et al., 2024) to the Related Work section (Q5 of Reviewer hSug).

We believe these revisions address the reviewers' main concerns and provide a clearer demonstration of the strengths of our proposed methods. Please let us know if you have any further questions or suggestions.

---

### Author Response · Authors · 2025-12-03
**Message to New Area Chairs on Rebuttals**

Dear Area Chairs,

We are highly appreciative of your assistance in managing our submission and the constructive feedback from the reviewers. We have carefully addressed all the comments in our revision and rebuttal. Below is a summary outlining the key points raised by the reviewers and the corresponding changes we have implemented.

The main concerns raised in the reviews include:

- interpretations of our theory, including the discussions of convergence rates (Reviewer J4sP), the improvement over prior works (Reviewer J4sP) and the parameter-free nature (Reviewer hSug), as well as potentially misleading introduction to Case 2 of Adam++ (Reviewer hSug); and the reason for nondecreasing $\eta_t$ and tuned base learning rate (Reviewer 6s59)
- trade-off in additional memory costs (Reviewer hSug)
- potential improvement by introducing ADOPT [1] (Reviewer hSug)
- issues on the ablation study, including absence of baselines in ablation study (Reviewer J4sP) and the discussions of the optimization effects vs the implicit regularization effects. (Reviewer 66vf)
- small typos in the original paper

To address these issues, we have:

- revised the manuscript to clarify the discussions on convergence rates, the advantage of our work and the , and the parameter-free essence of our proposed methods, and we have provided additional explanatory details for the proofs in Appendices B, C, and D and the explanation of Case 2 of Adam++;
- carefully discussed the intuition of nondecreasing $\eta_t$ and its importance in proof, the common practice of previous works on parameter-free methods for tuning some hyper-parameters, and the mitigation of additional memory costs when scaling up the size of models;
- attempted to introduce ADOPT algorithm and reported issues in our practice;
- added more experiments to validate the robustness of our proposed methods and the balance between the optimization effects and the implicit regularization effects;
- proofread the paper to fix all the typos.

While we have not received additional comments from Reviewers J4sP,  66vf or hSug, we are confident that our rebuttal and revision thoroughly addressed their concerns.

We kindly request that you take these points into consideration during your final assessment. We sincerely thank you once more for your time and diligent effort in overseeing the review process.

Best regards,

Authors



[1] Taniguchi, Shohei, et al. "ADOPT: Modified Adam Can Converge with Any with the Optimal Rate." Advances in Neural Information Processing Systems 37 (2024): 72438-72474.

---

### Meta-Review · Area_Chair_1X1h · 2026-01-06

**Summary:**

The paper proposes AdaGrad++ and Adam++, simple parameter-free variants of AdaGrad and Adam that adapt step sizes based on distance traveled, aiming to remove learning-rate tuning while retaining convergence guarantees. The main disagreement across reviewers is whether the theoretical guarantees are meaningful and clearly motivated, and whether the empirical evidence really supports the “parameter-free” claim beyond reduced overfitting or robustness tricks.

**Reviewer Concerns:**

1. Theory clarity and meaningfulness of guarantees. Several reviewers questioned whether the convergence rates are genuinely informative and well-motivated. Concerns include reliance on boundedness assumptions imported from prior work, unclear interpretation of constants, and whether the theory really explains the “parameter-free” behavior rather than just establishing convergence under auxiliary conditions. This was clarified but not fully convincing to all reviewers.
2. Novelty and positioning relative to prior parameter-free optimizers. Reviewers were split on whether the contribution goes beyond incremental simplification of existing ideas (DoG, D-Adaptation, AMSGrad variants). While simplicity and clean implementation were appreciated, it remained somewhat unclear whether this fills a real gap in the literature versus being a modest repackaging with proofs.
3. Empirical evidence vs interpretation (robustness, overfitting, adaptivity). Although additional experiments and ablations were added, some reviewers remained unconvinced that the observed gains reflect true parameter-free adaptivity rather than reduced overfitting or implicit regularization effects that could be matched by tuning baselines (e.g., warm-up or learning-rate schedules). The mechanism behind improved generalization was still debated.

**Reviewer Scores:**

Reviewer J4sP (2 > 2/3 likely):  Main theoretical and novelty concerns are clarified but not fully resolved; skepticism likely remains.
Reviewer 66vf (6 > 6 likely): Robustness experiments and added discussion directly address key concerns.
Reviewer hSug (4 > 5 likely): Added LM experiments, nonconvex analysis appendix, and ADOPT discussion help materially, even if not perfect.
Reviewer 6s59 (4 > 4 likely): Authors responded carefully, but reviewer follow-up shows lingering conceptual doubts about adaptivity and overfitting explanation.

---

### Decision · Program_Chairs · 2026-01-26

Reject